# Schrödinger's FP: Training Neural Networks with Dynamic Floating-Point Containers

## Abstract

We introduce a software-hardware co-design approach to reduce memory traffic and footprint during training with BFloat16 or FP32, in order to boost energy efficiency and execution time performance. Our methods dynamically adjust the size and format of the floating-point containers used to store activations and weights during training. The different value distributions lead us to different approaches for exponents and mantissas. *Gecko* exploits the favourable exponent distribution with a lossless delta encoding approach to reduce the total exponent footprint by up to $58\%$ in comparison to the FP32 baseline. To contend with the noisy mantissa distributions, we present two lossy methods to eliminate as many as possible least significant bits without affecting accuracy. *Quantum Mantissa* is a machine learning mantissa compression method that taps onto the gradient descent algorithm to *learn* the minimal mantissa bitlengths on a per-layer granularity, and obtain up to $92\%$ reduction in total mantissa footprint. Alternatively, *BitChop* observes changes in the loss function during training to adjust mantissa bitlength network-wide, yielding a reduction of $81\%$ in footprint. *Schrödinger's FP* implements hardware encoders/decoders that, guided by *Gecko/Quantum Mantissa* or *Gecko/BitChop*, transparently encode/decode values when transferring to/from off-chip memory, boosting energy efficiency and reducing execution time.

## 1 Introduction

Training most state-of-the-art neural networks has become an exascale class task (Venkataramani et al., 2017; Amodei et al., 2018) requiring many graphics processors (NVidia, 2017) or specialized accelerators, e.g., (Jouppi et al., 2017; Hab, 2019; Liao et al., 2019; Cer, 2019). While training is both computationally and data demanding, it is the memory transfers to off-chip DRAM for *stashing* (i.e., saving and much later recovering) activation and weight tensors that dominate overall execution time and energy (Jain et al., 2018) (see Fig. 1). The per batch data volume easily surpasses on-chip memory capacities, necessitating off-chip DRAM accesses which are up to two orders of magnitude slower and more energy expensive. It's no wonder that reducing this overhead has been receiving attention throughout the software/hardware stack.

Chen et al. (2016) and Zheng et al. (2020) recompute rather than stash activations, whereas micro-batching strives to keep activations on chip (Huang et al., 2018). Encoding methods target specific value patterns such as zeros (Rhu et al., 2018) or redundant spatial information (Evans et al., 2020), or exploit underlying properties of training for certain tensors, e.g., the outputs of ReLU or Pooling (Jain et al., 2018). These lossless and lossy encodings use fewer bits for stashed tensor content to reduce tensor volume. This also boosts the effective capacity of each node's main memory, which further reduces traffic during distributed training. All aforementioned methods either shift significant costs to compute or target only some values and offer only limited relief.

The most direct way to reduce tensor volume is to use a more compact datatype. Initially, with the goal to demonstrate that neural networks can tackle challenging problems, training relied on single precision 32b floating-point (FP32), which still remains the datatype of choice when achieving the best accuracy is the priority. Recently, we have seen *some* success in training with more compact datatypes such as half-precision FP16, BFloat16 (Kalamkar et al., 2019), dynamic floating-point (Das et al., 2018), and flexpoint (Köster et al., 2017) and even with using combinations with other datatypes such as fixed-point (Das et al., 2018; Micikevicius et al., 2018; NVIDIA; Drumond et al., 2018). IBM

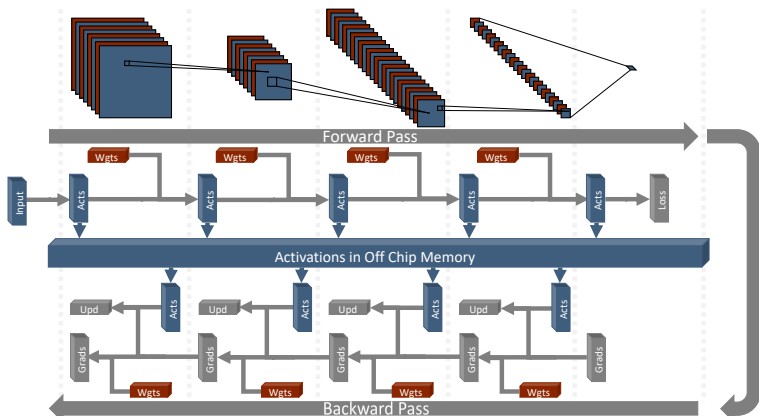

Figure 1: Training process and its memory transfers. Blue - Activations that are typically saved to off-chip memory during forward pass and retrieved during backward pass, Red - Weights that are typically stored and loaded once from off-chip memory, Gray - Updates and Gradients – through mini-batching during the backward pass they can often fit on-chip

managed to push the datatype to 8b (Wang et al., 2018b) and 4b (Sun et al., 2020) extremes for *some* cases. As Moore's law and Dennard scaling for semiconductors have come to an end, using more efficient datatypes during training is getting wider attention – even major hardware manufacturers are investigating how to use 8b floating point with different mantissa/exponent ratios according to perceived needs of tensors (Micikevicius et al., 2022). These methods require careful trial-and-error investigation of *where*, *when*, and *which* narrow datatypes to use. This is especially true because different tensors, tasks, architectures, or layers *require* different datatypes. Consequently, there is no guarantee of success. The methods require trial-and-error full training runs as whether the choice of datatypes is viable can only be evaluated *post mortem*. Moreover, since the datatypes are statically chosen they offer no opportunity to amend the choice if accuracy suffers (e.g., significant drop with deeper networks identified by IBM (Sun et al., 2020)).

Obviously, knowing in advance which compact datatypes to use during training would be the best. However, given that this goal still eludes us, our work asks whether we can harness the training process itself to automatically *learn* them. Ideally, such a method would *automatically* tailor datatypes to meet the demands of each tensor, layer, and network. Furthermore, it could continuously adjust datatype selection as training progresses, adapting to the changing needs. In addition to accelerating training, methods such as ours can further inform efforts for selecting more efficient datatypes for inference such as those by (Micikevicius et al., 2022) or (Sun et al., 2020).

A similar idea has successfully targeted fixed-point *inference* by using reinforcement learning (Wang et al., 2018a), clever differentiable datatype definitions (Nikolić et al., 2020), architecture search (Wu et al., 2018), and profiling (Nikolić et al., 2018), etc. However, all of these are too expensive for training and their overheads would overshadow the benefits of a more compact training datatype.

Given that floating point remains the datatype of choice, we focus on floating-point datatype selection. We explore the possibility to *dynamically* and *continuously* adjust the *mantissa bitlength* (fractional bits) and the *container* (overall bits) for floating-point values (activations and/or weights) for stashed tensors, and to do so *transparently* at no additional burden to the user. Our solution is *Schrödinger's FP*, a family of methods that dynamically adjust the floating-point encoding and complement the aforementioned training acceleration methods. Our approach is end-to-end fully-automated, requiring no input, guessing, or advanced knowledge from the operator. *Schrödinger's FP* can be used to reduce memory overheads and boost computation throughput. In this work, we limit our attention to boosting energy efficiency and performance by using *Schrödinger's FP* to transparently encode values as they are being stashed to off-chip memory, and decode them to their original format as they are being read back. This application can be used as a plug-in over any hardware without changing the existing on-chip memory hierarchy and compute units. Similarly, *Schrödinger's FP* will generally work in conjunction with methods that can improve accuracy for a preselected datatype, partition, distribute, or reschedule the training work to improve energy efficiency and performance.

*Schrödinger's FP* uses tailored approaches for the mantissa and exponent. It dynamically adjusts mantissa bitlengths in order to store and read fewer bits per number in off-chip memory. This work explores two such methods. The first, *Quantum Mantissa*, harnesses the training algorithm itself to learn on-the-fly the mantissa bitlengths that are needed per tensor/layer and continuously adapts those bitlengths per batch. *Quantum Mantissa* introduces a single learning parameter per tensor and a loss

function that includes the effects of the mantissa bitlength. Learning the bitlength incurs a negligible overhead compared to saving from the resulting reduction in off-chip traffic. The *Quantum Mantissa* experiments show that: 1) it reduces the mantissa bitlengths considerably, and 2) the reductions are achieved fairly soon in the training process and remain stable till the end. However, the bitlengths vary per tensor and fluctuate throughout, capturing benefits that wouldn't be possible with a static network-wide choice of datatype.

Motivated by the success of *Quantum Mantissa*, we explore the second mantissa adjustment method, *BitChop*, which requires no additional loss function and parameters. *BitChop* interface only needs to be notified of the per-batch updates to the loss. Using an exponential moving average of these changes, *BitChop* adjusts mantissa bitlength for the whole network. As long as the network seems to be improving, *BitChop* will attempt to use a shorter mantissa; otherwise, it will increase it. The method proves effective, albeit with lower reductions compared to *Quantum Mantissa*. This is expected since 1) *Quantum Mantissa* harnesses the training process to continuously learn the optimal bitlengths, and 2) *Quantum Mantissa* adjusts bitlengths per layer whereas *BitChop* uses a network-wide one.

Most of the exponents during training exhibit a heavily biased distribution (Awad et al., 2021). Accordingly, *Schrödinger's FP* uses a value-based approach that stores exponents using only as many bits as necessary to represent their magnitude and sign. Metadata encodes the number of bits used. To reduce the metadata overhead, *Schrödinger's FP* encodes exponents in groups.

To maximize benefits, we present a hardware-assisted implementation of *Schrödinger's FP*. The inclusion of specialized hardware units is now commonplace among all hardware vendors as the method of choice for further improving compute performance. Appendix A presents efficient hardware (de)compressors that operate on groups of otherwise unmodified floating-point values, be it FP32 or BFloat16. The compressors accept an external mantissa length signal and pack the group of values using the aforementioned compression methods for the mantissas and exponents. The decompressors expand such compressed blocks back into the original floating-point format. We demonstrate that *Schrödinger's FP* greatly improves energy efficiency and execution time. Our compression methods also serve as motivation for pursuing, as future work, a software-only implementation that would require low-level changes in the closed-source tensor operation kernels.

We highlight the following experimental findings:

- *Schrödinger's FP* compression techniques find the necessary mantissa and exponent bitlengths to reduce overall memory footprint without noticeable loss of accuracy: our *Quantum Mantissa*-based method reduces the tested models down to $20.8\%$ on average (range: $14.7\% - 24.9\%$) and our *BitChop*-based one to $24.6\%$ on average (range: $19.4\% - 28.9\%$)

- *Schrödinger's FP* compressor/decompressor exploit the reduced footprint to obtain $2.94\times$ and $2.64\times$ performance improvement for $\mathrm{S}FP_{\mathrm{QM}}$ and $\mathrm{S}FP_{\mathrm{BC}}$ (*Schrödinger's FP* with *Quantum Mantissa* or *BitChop*, see Section 2.1.4 for details), respectively. In fact, we hit a hard performance boundary, since all layers are shifted from memory bound to compute bound, which completely flips the workload paradigm.

- Crucially, *Schrödinger's FP* excels at squeezing out energy savings with on average, $3.38\times$ and $2.96\times$ better energy efficiency for $\mathrm{S}FP_{\mathrm{QM}}$ and $\mathrm{S}FP_{\mathrm{BC}}$.

## 2 ADJUSTING VALUE CONTAINERS DURING TRAINING

Generally, maintaining accuracy on most real-world tasks requires training with a floating-point approach. Floating-point formats comprise three segments: a mantissa, an exponent, and a sign bit. Mantissas and exponents are differently distributed, necessitating different approaches. The greatest challenge is compressing mantissas since they are uniformly distributed across the domain, whereas compression exploits non-uniformity. We will present two methods to compress mantissas, a machine learning approach (Section 2.1) and a hardware-design-inspired approach (Section 2.2). In contrast, exponents can be compressed with fairly simple hardware techniques (Section 2.3).

We study *Schrödinger's FP* with ResNet18 and ResNet50 (He et al., 2015), and MobileNet V3 (Howard et al., 2019) trained on ImageNet (Russakovsky et al., 2014), DLRM (Naumov et al., 2019) trained on Kaggle Criteo, BERT (Devlin et al., 2018) finetuned on MRPC (Dolan & Brockett,

2005) and GTP–2 (Radford et al., 2019) finetuned on Wikitext 2 (Merity et al., 2016). For clarity, we report detailed results with ResNet18 with BFloat16 throughout the paper, concluding with overall performance and energy efficiency measurements for all models.

## 2.1 MANTISSA: *Quantum Mantissa*

*Quantum Mantissa* involves procedures for both the forward and backward pass of training. We begin by defining a conventional quantization scheme for integer mantissa bitlengths in the forward pass, and then expand it to the non-integer domain, and describe how this interpretation allows bitlengths to be learned using gradient descent. Subsequently, we introduce a parameterizable loss function, which enables *Quantum Mantissa* to penalize larger bitlengths. We then briefly touch on the compute overhead of our method and the plan for final selection of mantissa bitlengths. Ultimately, we demonstrate the benefits of *Quantum Mantissa* on memory footprint during ImageNet training.

### 2.1.1 QUANTIZATION

The greatest challenge for learning bitlengths is that they represent discrete values with no obvious differentiation. To overcome this, we define a quantization method on non-integer bitlengths. We start with an integer quantization of the mantissa $M$ with $n$ bits by zeroing out all but the top $n$ bits:

$$Q(M, n) = M \wedge (2^n - 1) << (m - n) \tag{1}$$

where $Q(M, n)$ is the quantized mantissa with bitlength $n$, $m$ is the maximum number of bits and $\wedge$ represents bitwise AND.

Throughout training, we represent the integer quantization as $Q(M, n)$. This scheme does not allow the learning of bitlengths with gradient descent due to its discontinuous and non-differentiable nature. To expand the definition to real-valued $n = \lfloor n \rfloor + \{n\}$, the values used in inference during training are stochastically selected between the nearest two integers with probabilities $\{n\}$ and $1 - \{n\}$:

$$Q(M, n) = \begin{cases} Q(M, \lfloor n \rfloor), & \text{with probability } 1 - \{n\} \\ Q(M, \lfloor n \rfloor + 1), & \text{with probability } \{n\} \end{cases} \tag{2}$$

where $\lfloor n \rfloor$ and $\{n\}$ are floor and fractional parts of $n$, respectively. The scheme can be, and in this work is, applied to activations and weights separately. Since the minimum bitlength per value is 0, $n$ is clipped at 0. This presents a reasonable extension of the meaning of bitlength in continuous space and allows for the loss to be differentiable with respect to bitlength.

During the forward pass, the formulae above are applied to both activations and weights. The quantized values are saved and used in the backward pass. During the backward pass, we use the straight-through estimator (Bengio et al., 2013; Hubara et al., 2016) to prevent propagating zero gradients that result from the discontinuity's discreteness; however, we use the quantized mantissas for all calculations. This efficient quantization during the forward pass reduces the footprint of the whole process.

### 2.1.2 LOSS FUNCTION

On top of finding the optimal weights, the modified loss function penalizes mantissa bitlengths by adding a weighted average (with weights $\lambda_i$, not to be confused with the model's weights) of the bits $m_i$ required for mantissas of weights and activations. We define total loss $L$ as:

$$L = L_l + \gamma \sum^i (\lambda_i \times n_i) \tag{3}$$

where $L_l$ is the original loss function, $\gamma$ is the regularization coefficient used for selecting how aggressive the quantization should be, $\lambda_i$ is the weight corresponding to the importance of the $i^{th}$ group of values (one per tensor), and $n_i$ is the bitlength of the activations or weights in that tensor.

This loss function can be used to target any quantifiable criteria by a suitable selection of the $\lambda_i$ parameters. Since our goal is to minimize the total footprint of a training run, we weigh each layer's tensors according to their memory footprint.

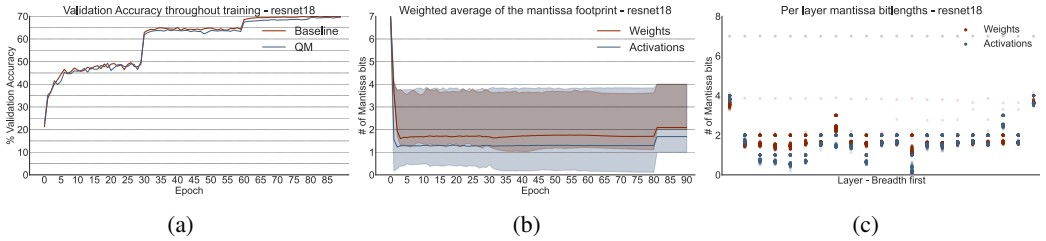

Figure 2: *Quantum Mantissa* on ResNet18/ImageNet: (a) Validation accuracy throughout training, (b) Weighted mantissa bitlengths with their spread throughout training, and (c) Mantissa bitlengths for each layer at the end of each epoch. Darker dots represent the latter epochs.

### 2.1.3 Computational and Memory Overheads

*Quantum Mantissa* adds minimal computational and memory overhead to the forward and backward passes. In the forward pass, random numbers need to be created at a chosen granularity to determine the quantized values. Our experiments show that per tensor/layer is sufficient and is a negligible cost.

To update the bitlength parameters in the backward pass, we need to compute their gradients. These are a function of the weight values and gradients, which are calculated during the regular backward pass. The extra calculations for each bitlength are proportional to the number of values quantized to that bitlength. This overhead is negligible in comparison to the total number of computations. For our experiments, the overhead is less than $2\%$ and $0.5\%$ for MobileNet V3 and ResNet18, respectively.

On the memory side, the only extra parameters that need to be saved are the bitlengths, two floats per layer (bitlength for weights and activations), again negligible in comparison with the total footprint. All other values are consumed as they are produced without need for off-chip stashing.

### 2.1.4 Bitlength Selection

*Quantum Mantissa* will produce non-integer bitlengths and requires non-deterministic inference. We prefer the deployed network not to have this requirement. For this reason, we *round up* the bitlengths and fix them for some training time to fine-tune the network to this state. While our experiments show that bitlengths converge quickly and final ones can be determined within a couple of epochs, avoiding the small overhead for most of the training, we delay this action so that bitlengths have the ability to increase if needed during training. Our experiments show that this is unnecessary for the models studied; however, the overhead is so small that we leave it on as a safety mechanism. We round up the bitlengths for the last 10 epochs to let the network regain any accuracy that might have been lost due to *Quantum Mantissa*. *Quantum Mantissa* still reduces traffic during these epochs.

**Evaluation: BitLengths and Accuracy**  We report measurements for per-layer weights and activations quantized separately using a loss function weighted to minimize total memory footprint. We train ResNet18 on the ImageNet dataset over 90 epochs, with regularizer strength of 0.1, learning rate of 0.1, 0.01 and 0.001 respectively at epochs 0, 30, and 60 and weight decay of 0.0001.

*Quantum Mantissa* excels at minimizing the memory footprint whilst not introducing accuracy loss. Figure 2a shows that throughout training, *Quantum Mantissa* introduces minimal changes in validation accuracy. In the end, we converge to a solution within $0.4\%$ of the FP32 baseline.

Figure 2b shows how *Quantum Mantissa* quickly (within a couple of epochs) reduces the required mantissas for activations and weights down to $1 - 2$ bits on average. Throughout training, the total cumulative memory footprint is reduced to $7.8\%$ and $25.5\%$ of the FP32 and BFloat16 mantissa footprint, respectively. The figure further shows that there is a large spread across different layers, indicating that a granular, per-layer, approach is the right choice to maximize benefits. Via the weighted loss function, *Quantum Mantissa* generally targets the activation bitlengths more aggressively than the weights because the activations are responsible for the majority of the memory footprint.

The spread of mantissa bitlengths across the network and time is shown in Figure 2c. While most layers quickly settle at 1 or 2 bits, there are exceptions that require more (up to 4b). Consequently, a network-scale datatype would have to use the largest one and leave a lot of the potential untapped. For ResNet18, the maximum bitlength is over $2\times$ larger than the weighted average.

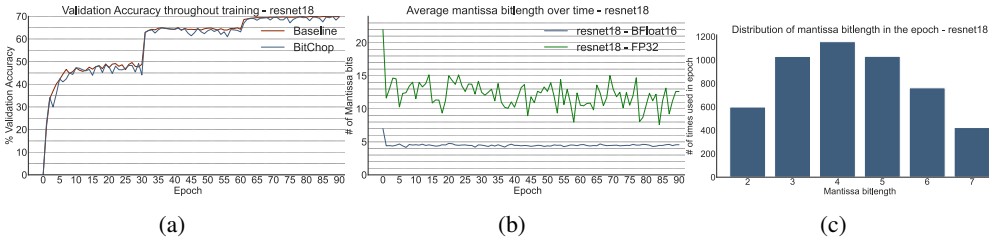

Figure 3: *BitChop* on ResNet18/ImageNet: (a) Validation accuracy throughout BFloat16 training, (b) Average mantissa bitlengths per epoch throughout training, on BFloat16 and FP32, (c) Distribution of BitChop's mantissa bitlengths throughout the 5005 batches of epoch 45 over BFloat16.

## 2.2  MANTISSA: *BitChop*

While *Quantum Mantissa* leverages training itself to greatly reduce mantissa lengths, having a method that does not require introducing an additional loss function and parameters is appealing. *BitChop* is a run-time, heuristic method to reduce the number of mantissa bits for the forward and backward passes. At a high-level, *BitChop* monitors how training progresses adjusting the mantissa length accordingly: as long as the training seems to be improving the network, *BitChop* will attempt to use a shorter mantissa; otherwise, it will try to increase its bitlength. *BitChop* conceptually splits the training process into *periods*, where a period is defined as processing $N$ batches. *BitChop* adjusts the mantissa at the end of each period using information on network progress.

The ideal scenario for *BitChop* is one where past observation periods are good indicators of forthcoming behavior. Macroscopically, accuracy improves during training which appears to be a good fit. Microscopically, however, training is a noisy process. Fortunately, training is a relatively long process based on "trial-and-error" which may be forgiving for momentary lapses in judgement.

There are three major design decisions that impact how successful *BitChop* can be: 1) what information to use as a proxy for network progress, 2) how long the period should be, and 3) at what granularity to adjust mantissa lengths. The resulting method should strike a balance between capturing as much as possible the opportunity to reduce bitlength, and avoiding over-clipping as this can hurt learning progress and ultimately the final accuracy that would be achieved.

We have experimented with several options and arrived at the following choices: 1) Using an exponential moving average of the loss as a proxy for network progress, and 2) using a short period where $N = 1$, that is a single batch. Additionally, rather than attempting to adjust mantissas at the tensor/layer level, *BitChop* uses the same mantissa for the whole model. Specifically, to monitor network progress, *BitChop* uses the loss which is calculated per batch as part of the regular training process. While the loss improves over time, when observed over short periods of time, it exhibits non-monotonic behavior which is sometimes erratic. To compensate for this volatility, *BitChop* uses an exponential moving average $Mavg$ which it updates at the end of each period:

$$Mavg = Mavg + \alpha * (L_i - Mavg) \tag{4}$$

where $L_i$ is the loss during the last period and $\alpha$ is an exponential decay factor which can be adjusted to assign more or less significance to older loss values. This smooths the loss over time while giving importance to the most recent periods. At the end of each period $i$, *BitChop* adjusts the mantissa bitlength (unchanged, lower, or higher) by comparing $L_i$ with $Mavg$ within a dynamically updated threshold $T$.

**Evaluation: Bitlengths and Accuracy**  We report *BitChop*'s effect on activation footprint and accuracy during full training sessions of ResNet18 as before. Figure 3a shows that the network achieves the same validation accuracy as with the baseline training. For clarity, the figure shows results for BFloat16 only (results with FP32 were similar and accuracy was unaffected). Throughout the training process, validation accuracy under *BitChop* exhibits more pronounced swings compared to the baseline and to *Quantum Mantissa*. However, in absolute terms, these swings are small.

Figure 3b shows that *BitChop* reduces mantissa bitlengths to 4 - 5 bits *on average* when used over BFloat16 and to 12 bits on average when used over FP32. However, mantissa bitlengths vary per batch depending on the loss as illustrated in the histogram (Figure 3c) of the bitlengths used throughout a sample epoch (epoch 45) for the BFloat16 run. This shows that the training process sometimes requires the entire range of Bfloat16 whereas other times it only requires 2 bits. All across the training process, *BitChop* reduces the total mantissa footprint of the BFloat16 baseline to $64.3\%$. Over

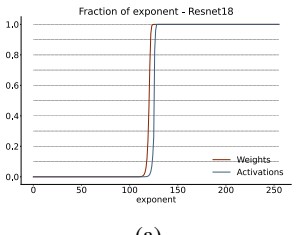 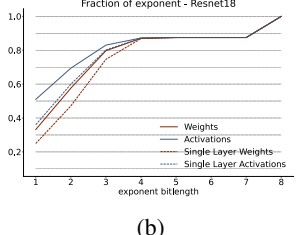

(a)                                            (b)

Figure 4: *Gecko* on ResNet18/ImageNet: (a) Cumulative distribution of exponent values. (b) Post-encoding cumulative distribution of exponent bitlength

FP32 *BitChop* reduces mantissa footprint to $52.3\%$. While *BitChop* might miss potential bitlength reductions, it is non-intrusive and has virtually no overhead.

## 2.3   EXPONENT: *Gecko*

The exponents of BFloat16 and FP32 are 8b biased integers. Except for a few early batches, we find that during training, the exponent values exhibit a heavily biased distribution centered around 128 which represents 0. This is illustrated in Figure 8a which reports the exponent distribution throughout training of ResNet18 after epoch 10. We omit gradients which are even more biased as those can be kept on-chip. Taking advantage of the relatively small magnitude of most exponents, we adopt a variable length *lossless* exponent encoding. The encoding uses only as many bits as necessary to represent the specific exponent magnitude rather than using 8b irrespective of the value. Due to our variable-sized exponents, a 3b metadata field specifies the number of bits used. Having a dedicated bitlength per value would negate any benefits or worse, become an overhead. To amortize this cost, multiple exponents share a common bitlength that is long enough to accommodate the largest one within the group. We further observe that, especially for weights, the values exhibit spatial correlation (values that are close by have similar magnitude). Encoding differences in value skews the distribution closer to zero, benefiting our approach.

The specific encoding scheme *Gecko* used is as follows: Given a tensor, *Gecko* first groups the values in groups of 64 (padding as needed) which it treats conceptually as an 8x8 matrix. Every column of 8 exponents is a group which shares a common base exponent. The base exponent per column is the exponent that appears in the first row of incoming data. The base exponent is stored in 8b. The remaining 7 exponents are stored as deltas from the base exponent. The deltas are stored as [magnitude, sign] format and using a bitlength to accommodate the highest magnitude among those per row. A leading 1 detector determines how many bits are needed. The bitlength is stored using 3b and the remaining exponents are stored using the bitlength chosen.

**Evaluation: BitLength**  We measure how many bits are needed to encode the exponents using *Gecko* for the duration of training of ResNet18 as described previously. As representative measurements, Figure 8b reports the cumulative distributions of exponent bitlength for one batch across 1) all layers, and 2) for a single layer, separately for weights and activations. After delta encoding, almost 90% of the exponents are lower than 16, and 20% of the weight exponents and 40% of the activation exponents need only 1 bit. Across the whole training process, the overall compression ratio for the weight exponents is $0.56$ and $0.52$ for the activation exponents. The ratio is calculated as $(M + C)/O$ where $M$ the bits used by the per group bitlength fields, $C$ the bits used to encode the exponent magnitudes after compression, and $O$ the bits used to encode exponents in the original format.

## 3   EVALUATION – PUTTING ALL TOGETHER

We study the following two *Schrödinger's FP* variants: *Gecko* with *Quantum Mantissa* ($\mathrm{S}FP_{\mathrm{QM}}$) and *Gecko* with *BitChop* ($\mathrm{S}FP_{\mathrm{BC}}$) which are combinations of our exponent and mantissa compression methods and the interaction with GIST++ (Jain et al., 2018). GIST++ is a slightly modified version of Gist that uses sparsity encoding only for those tensors where doing so reduces the footprint, avoiding the increase in traffic that would occur otherwise. For instance, this is useful for MobileNet V3, BERT and GPT-2 which do not use ReLU, and as a result, exhibit very little sparsity.

We perform *full* training for ResNet18, ResNet50 and MobileNet V3 Small on ImageNet, DLRM on Kaggle Criteo as well as finetuning BERT on MRCP and GPT-2 on Wikitext 2, using an RTX3090/24GB with PyTorch v1.10. We implement *Quantum Mantissa* by modifying the loss

Table 1: $SFP_{BC}$, $SFP_{QM}$, BF16: Val. Accuracy/Perplexity and total memory reduction vs. FP32.

| Network | Task | Metric | FP32 Score | FP32 Footprint | BF16 Footprint | $SFP_{QM}$ Score | $SFP_{QM}$ Footprint | $SFP_{BC}$ Score | $SFP_{BC}$ Footprint |
|---|---|---|---|---|---|---|---|---|---|
| **ResNet18** | Classification | Accuracy | 69.94 | 100% | 50% | 69.54 | 14.7% | 69.95 | 23.7% |
| **ResNet50** | Classification | Accuracy | 76.06 | 100% | 50% | 75.58 | 20.6% | 75.72 | 21.7% |
| **MobileNet V3 S** | Classification | Accuracy | 65.60 | 100% | 50% | 65.26 | 24.9% | 65.21 | 27.2% |
| **BERT** | Text classification | Accuracy | 84.56 | 100% | 50% | 84.31 | 17.9% | 84.42 | 19.4% |
| **GPT-2** | Language Modeling | Perplexity | 20.95 | 100% | 50% | 20.96 | 23.5% | 20.95 | 28.9% |
| **DLRM** | Recommendation | Accuracy | 79.42 | 100% | 50% | 79.45 | 23.1% | 79.44 | 26.9% |

function and adding the gradient calculations for the per tensor/layer parameters. We simulate *BitChop* in software. For both methods, we faithfully emulate mantissa bitlength arithmetic effects by truncating the mantissa bits at the boundary of each layer using PyTorch hooks and custom layers. We also measure *Gecko*'s effects in software via PyTorch hooks. The above enhancements allow us to measure the effects our methods have on traffic and model accuracy.

## 3.1 MEMORY FOOTPRINT REDUCTION

First we report activation and weight footprint reduction on ResNet18. Table 1 shows the cumulative total memory reduction and validation accuracies in comparison with FP32 and BFloat16 baselines. Combined, our compression techniques excel at reducing footprint, with little affect on accuracy.

$SFP_{QM}$: Figure 5a shows the relative footprint of each part of the datatype with $SFP_{QM}$ in comparison with the FP32 and Bfloat16 baseline. Even though our methods are very effective at reducing the weight footprint ($91\%$ for mantissas and $54\%$ for exponents), this effect is negligible in the grand scheme of things due to the fact that weights are a very small part of all three footprints. For the same reason, the reductions in activation footprint ($92\%$ for mantissas, $63\%$ for exponents and $98\%$ for sign) have a far greater effect. Because of the effectiveness of *Quantum Mantissa*, the mantissas are reduced from the top contributor in FP32 ($70\%$), to a minor contributor ($38\%$). While exponents are significantly reduced too, they start to dominate with $59\%$ of the footprint in comparison with FP32 at $24\%$. Similar conclusions are reached when comparing with Bfloat16 except for the fact that mantissas and exponents start with similar footprint.

$SFP_{BC}$: Figure 5a also shows the relative footprint of the datatype components under $SFP_{BC}$ when compared to the FP32 and Bfloat16 baselines. While *BitChop* does reduce mantissa bitlength for the network's weights, this does not have a great effect in the total memory footprint reduction due to the small size of weights when compared to activations. Although mantissa weight footprint is not reduced, weight exponent footprint is by $56\%$. This is why the focus on the activations' mantissa bitlengths yields a significant total memory footprint reduction when compared to FP32 (mantissa footprint is reduced by $81\%$, exponent footprint by $63\%$ and sign by $98\%$ in activations), and a smaller but still significant reduction when compared with Bfloat16 ($36\%$ for mantissa and $63\%$ for exponents). The reductions are not as great as with *Quantum Mantissa* due to the network-wise limitation of the method and activation mantissas stay as the major contributor of footprint.

## 3.2 RELATIVE COMPRESSION AND COMPARISON WITH OTHER METHODS

Finally, we compare *Schrödinger's FP* compression against Bfloat16, GIST++ and JS, a simple sparse Bfloat16 zero-compression method in Figure 5b. JS uses an extra bit per value to avoid storing zeros. We limit attention to activations since weights represent a small fraction of the overall footprint and traffic. All methods benefit from using 16b. On ResNet18, JS and GIST++ benefit from the $30\%$ reduction due to high sparsity induced by ReLu. GIST++ benefits even further because of its efficient compression of maximum pooling. $SFP_{BC}$ does even better just by finding a smaller datatype outperforming all of them, whereas $SFP_{QM}$ proves even better by adjusting the datatype per layer. However, $SFP_{BC}$ and $SFP_{QM}$ only target the reduced datatype and there is an opportunity to build on it with the same ideas that power JS and GIST++. When combined, this further improves compression ratios to $10\times$ and $8\times$ for modified $SFP_{QM}$ and $SFP_{BC}$.

MobileNet V3 Small poses a bigger challenge since it sparsely uses ReLu and uses no max pooling. Accordingly, there is little potential for JS and GIST++ to exploit. $SFP_{QM}$ and $SFP_{BC}$ still get another $2\times$ compression over Bfloat16, JS, and GIST++. Application of ideas from JS and GIST++ to *Schrödinger's FP* compression offers only marginal gains. BERT and GPT-2 will exhibit similar results due to not using ReLu.

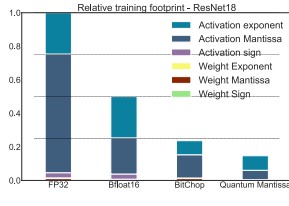 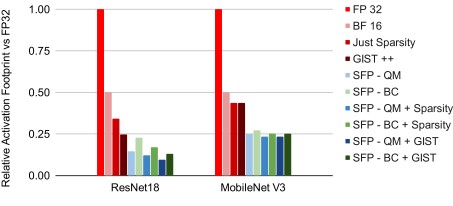

(a)                                             (b)

Figure 5: *Schrödinger's FP*: a) Relative training footprint ResNet18/ImageNet w/ FP32, BF16, S$FP_{BC}$ and S$FP_{QM}$, and b) Cumulative activation footprint w/ BF16, sparsity only and GIST++.

Table 2: Performance and Energy Efficiency gains in comparison w/ FP32

| Network | Performance | | | Energy Efficiency | | |
|---|---|---|---|---|---|---|
| | **Bfloat 16** | $SFP_{QM}$ | $SFP_{BC}$ | **Bfloat 16** | $SFP_{QM}$ | $SFP_{BC}$ |
| **ResNet18** | 1.53× | 2.30× | 2.09× | 2.00× | 6.12× | 4.22× |
| **ResNet50** | 1.64× | 2.53× | 2.45× | 1.98× | 4.56× | 4.48× |
| **MobileNet V3 S** | 1.72× | 2.37× | 2.14× | 2.00× | 3.95× | 3.60× |
| **BERT** | 1.88× | 3.77× | 3.54× | 1.36× | 1.78× | 1.73× |
| **GPT-2** | 2.13× | 4.48× | 3.42× | 1.34× | 1.70× | 1.58× |
| **DLRM** | 1.60× | 2.19× | 2.22× | 1.58× | 2.14× | 2.17× |

## 3.3 PERFORMANCE AND ENERGY EFFICIENCY

We evaluate the execution time and energy efficiency by incorporating *Gecko* hardware units into a hardware accelerator representative of state-of-the-art designs. We model an accelerator with 8K units (each capable of performing 4 MACs per cycle), and a 500MHz clock for a peak computer bandwidth of 16TFLOPS. We consider two baseline configurations using respectively FP32 and BFloat16. Both have 8 channels of LPDDR4-3200 DRAM memory and 32MB of on-chip buffers. Appendix B details the evaluation methodology. Area overhead of the compressor and decompressor is 0.67% of accelerator area, excluding on-chip memory, which is negligible.

Table 2 reports execution time improvements of Bfloat16, $SFP_{QM}$, and $SFP_{BC}$ over the FP32 baseline. On average, $SFP_{QM}$ and $SFP_{BC}$ are 2.9× and 2.6× faster respectively, compared to 1.8× with Bfloat16. Both $SFP_{QM}$ and $SFP_{BC}$ significantly outperform both the FP32 baseline and Bfloat16. However, performance does not scale linearly even though $SFP_{QM}$ and $SFP_{BC}$ reduce the memory footprint to 20.8% and 24.6% respectively: some layers that were previously memory bound during the training process become compute bound because of the reduction in memory footprint. This is the reason why even though Bfloat16 reduces the datatype to half, it does not achieve 2× speedup. This transition of most layers from memory bound to compute bound also affects the improvements in performance that $SFP_{QM}$ can offer, as even though it consistently achieves a lower footprint than $SFP_{BC}$, this only offers an advantage for performance in the few layers that remain memory bound. $SFP_{QM}$ may offer bigger performance benefits if coupled with higher computational performance hardware. Regardless, while a reduction in traffic may not yield a direct improvement in performance, it does improve energy efficiency.

Table 2 also shows energy efficiency improvement with Bfloat16, $SFP_{QM}$ and $SFP_{BC}$ over the FP32 baseline. $SFP_{QM}$ and $SFP_{BC}$ excel at improving energy efficiency by significantly reducing DRAM traffic. Since the energy consumption of DRAM accesses greatly exceeds that of computation, and some layers are or become compute bound, $SFP_{QM}$ and $SFP_{BC}$ improve energy efficiency more than performance, achieving an average of 3.4× and 3.0× energy efficiency respectively. The dominance of DRAM access energy consumption over computation can also be seen in Bfloat16, where the reduction to half the footprint, the use of 16-bit compute units, and the compute layers being no longer a limiting factor gives Bfloat16 a 1.7× energy efficiency.

## 4 CONCLUSION

We explored methods that dynamically adapt the bitlengths and containers used for floating-point values during training. The different distributions of the exponents and mantissas led us to tailored approaches for each. We target the largest contributors to off-chip traffic during training for both activations and weights. There are several directions for improvements and further exploration including expanding the methods to also target the gradients and refining the underlying policies they use to adapt mantissa lengths. Regardless, this work has demonstrated that the methods are effective. The key advantages of our methods are: 1) they are dynamic and adaptive, 2) they do not modify the training algorithm, and 3) they take advantage of value content for the exponents.

## 5 REPRODUCIBILITY STATEMENT

We will release our full code with all the necessary instructions on how to re-run our experiments by the camera-ready deadline at the latest.

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

## A  Our Hardware Approach

This section presents the *Schrödinger's FP* hardware encoder/decoder units that efficiently exploit the potential created by our quantization schemes. Without the loss of generality we describe compressors/decompressors that process groups of 64 FP32 values.

***Compressor:***  Figure 6a shows that the compressor contains 8 packer units (Figure 6c). The compressor accepts one row (8 numbers) per cycle, for a total of 8 cycles to consume the whole group. Each column is treated as a subgroup whose exponents are to be encoded using the first element's exponent as the base and the rest as deltas. Accordingly, the exponents of the first row are stored as-is via the packers. For the every subsequent row, the compressor first calculates deltas prior to passing them to the packers.

The length of the mantissa is the same for all values and is provided by the mantissa quantizer method be it *Quantum Mantissa* or *BitChop*. Each row uses a container whose bitlength is the sum of the mantissa bitlength (provided externally) plus the bitlength needed to store the highest exponent magnitude cross the row. To avoid wide crossbars when packing/unpacking, values remain within the confines of their original format bit positions as per the method proposed in Proteus (Judd et al., 2016). In contrast to Proteus, however, here every row uses a different bitlength, the values are floating-point, the bitlengths vary during runtime and per row, and we target training. The exponent lengths need to be stored as metadata per row. These are stored separately necessitating two write streams per tensor both however are sequential thus DRAM-friendly. The mantissa lengths are either tensor/layer- or network-wide and are stored along with the other metadata for the model.

Each packer (Figure 6c) takes a single FP32 number in [exponent, sign, mantissa] format, masks out unused exponent and mantissa bits, and rotates the remain bits to position to fill in the output row. The mask is created based on the exp_width and man_width inputs. The rotation counter register provides the rotation count which is updated to (exp_width+man_width+1) every cycle. The (L,R) register pair, is used to tightly pack the encoded values into successive rows. There are needed since a value may now be split across two memory rows. When either register, its 32b (or 16b for BFloat16) are drained to memory. This arrangement effectively packs the values belonging to this column tightly within a column of 32b in memory. Since each rows the same total bitlength, the 8 packers operate in tandem filling their respective outputs at exactly the same rate. As a result, the compressor produces 8x32b at a time. The rate at which the outputs are produced depends on the compression rate achieved, the higher the compression, the lower the rate.

***Decompressor:***  As Figure 6b shows, the decompressor mirrors the compressor. It takes 8 3-bit exponent widths and a mantissa length from the system, and 8x32 bits of data per cycle. Every column of 32b is fed into a dedicated unpacker per column. The unpacker (Figure 6d reads the exponent length for this row and the global mantissa length, takes the correct number of bits, and extends the data to [exponent, sign, mantissa] format.

Each unpacker handles one column of 32b from the incoming compressed stream. The combine-and-shift will combine the input data and previous data in register then shift to the left. The number of shifted bits is determined by the exponent and mantissa lengths of this row. The 32-bit data on the left of the register are taken out and shifted to the right (zero extending the exponent). Finally, the unpacker reinserts the mantissa bits that were trimmed during compression. Since each row of data uses the same total bitlength, the unpackers operate in tandem consuming data at the same rate. The net effect is that external memory see wide accesses on both sides.

## B  Hardware Evaluation Methodology

Best practices for the evaluation of custom hardware architectures necessitates exploration and validation first via analytical modelling or via cycle-accurate simulation. Since training these networks takes several *days* on actual hardware, cycle-accurate simulation of the full process is impractical. To estimate performance and energy, we use the best practice approach by analytically modelling the time and energy used per layer per pass of a baseline accelerator. To do so, we use traffic and compute counts collected during the aforementioned full training runs. We record these counts each time a layer is invoked using PyTorch hooks. We model time and energy for memory accesses via DRAMSIM3 (Li et al., 2020). For modeling on-chip structures we use CACTI (HewlettPackard) for the buffers and

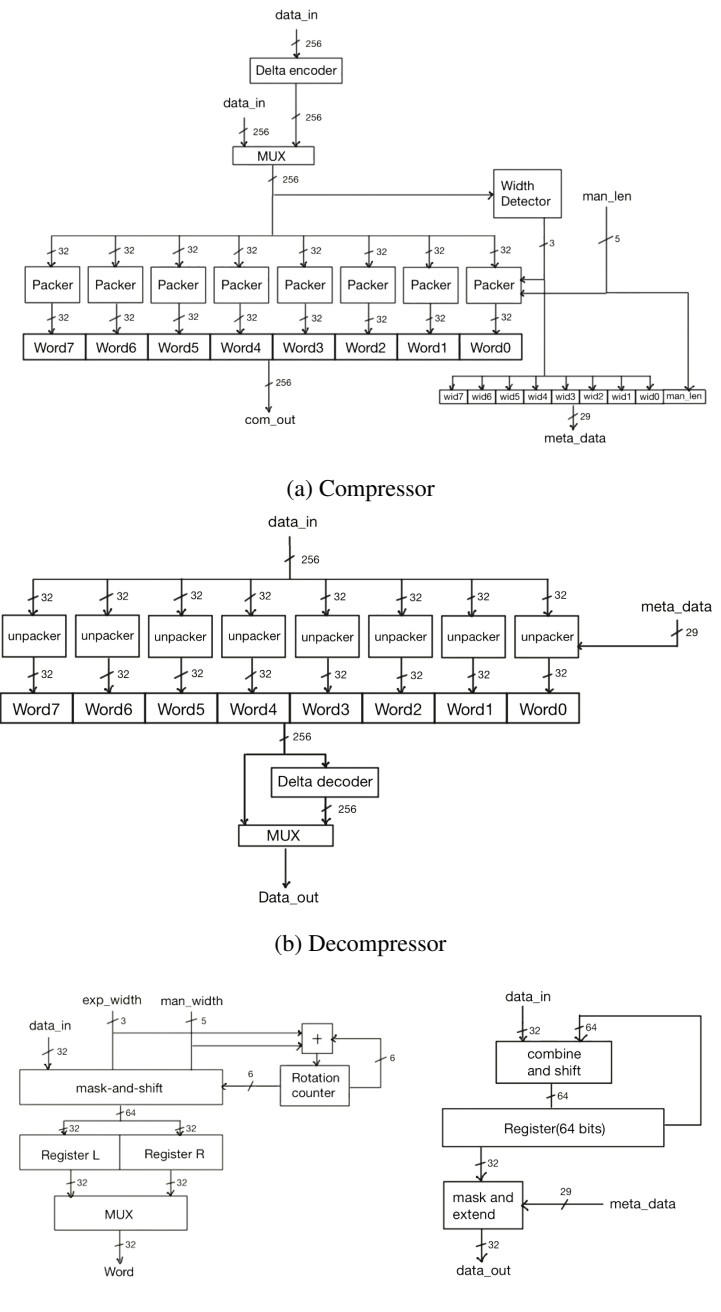

Figure 6: *Schrödinger's FP* Compressors/Decompressors

| module | area per unit ($um^2$) | unit number | total area ($mm^2$) |
|---|---|---|---|
| compressor | 40682.88 | 16 | 0.651 |
| decompressor | 46481.40 | 16 | 0.744 |
| accelerator | 38533.68 | 8000 | 308.27 |

Table 3: Hardware Area Overhead

layout measurements for the compute units and the *Gecko* compressors/decompressors. We use a commercial 65nm process to model the processing units and *Gecko* hardware. We implement the units in Verilog and perform synthesis via the Synopsys Design Compiler and *layout* using Cadence Innovus with a target frequency of 500MHz. Synthesis uses Synopsys' commercial Building Block IP library for the target tech node. We estimate power via Innovus using traces over a representative input sample to model properly signal activity. We used nominal operating conditions to model power and latency. There are two *Gecko* compressor/decompressor units per channel.

Due to the complexity and time cost of cycle-accurate hardware simulation, we have opted for an estimated time and energy consumption analytical model based on the proposed hardware description and the compressor-decompressor architecture. To compute the analytical model, we first analyze the network and retrieve its structure (layer input and output sizes, kernel sizes for convolutional layers, stride, bias and padding). We then calculate the compute operations that will happen for the general batch size (N) in both the forward and backward pass, as well as the number of parameters that must be stored in memory for activations, weights and gradients.

To take advantage of data reuse where possible we perform the forward pass in a layer-first order per batch. This allows us to read the weights per layer only once per batch. For the backward pass, we utilize the on-chip buffers for mini-batching with a layer-first order over a mini-batch of samples. Mini-batching reduces overall traffic by processing as many samples as possible in a layer-first order avoiding either having to spill gradients or reading and writing weights per sample per layer. The number of samples that can fit in a mini-batch depends on the layer dimensions and the size of the on-chip buffer.

Both $SFP_{QM}$ and $SFP_{BC}$ sample bitlengths per batch to a log file for both mantissas and exponents. These bitlengths are used to compute the number of mini-batches that can fit at every training step per layer on chip. Based on the number of sampled mini-batches (K) we compute the memory footprint generated on the forward pass for each method. After this, we calculate the footprint that stays on-chip and can be loaded from on-chip for the backward pass, and the footprint that goes to off-chip and has to be loaded to on-chip again for it. Based on these memory accesses, we use DRAMsim to simulate the number of compute-cycles that take the memory accesses to finish and we use the maximum cycles between compute and memory as the time constraint to calculate total computation time in the proposed hardware.

To calculate energy consumption and efficiency, we use the information gathered in terms of on-chip memory access cycles, off-chip memory access cycles and compute cycles. We estimate energy consumption for all components including the compressors and decompressors. We use the following equations to estimate energy consumption for our methods (all symbols are defined in table 5):

$$
\begin{aligned}
E_{forward} = E_{compute\,fwd} + E_{offchip\,in\,act\,mem} + \\
E_{offchip\,wgt\,mem} + E_{offchip\,out\,act\,mem} + E_{onchip\,in\,act\,mem} + \\
E_{onchip\,wgt\,mem} + E_{onchip\,out\,act\,mem} + E_{read\,ops\,mem} + \\
E_{decomp\,act} + E_{decomp\,wgt} + E_{comp\,act}
\end{aligned} \tag{5}
$$

$$
\begin{aligned}
E_{backward} = E_{compute\,bck} + E_{offchip\,in\,act\,mem} + \\
E_{offchip\,wgt\,mem} + E_{onchip\,in\,act\,mem} + \\
E_{onchip\,wgt\,mem} + E_{read\,ops\,mem} + \\
E_{decomp\,act} + E_{decomp\,wgt}
\end{aligned} \tag{6}
$$

where,

| Compression ratio | Compressor power (mW) | Decompressor power (mW) |
|---|---|---|
| 0.143 - 0.263 | 10.87 | 13.84 |
| 0.264 - 0.388 | 12.18 | 14.72 |
| 0.389 - 0.513 | 12.65 | 15.97 |
| 0.514 - 0.638 | 13.44 | 15.76 |
| 0.639 - 0.763 | 14.98 | 15.42 |

Table 4: $P()$ terms: Power consumption as a function compression ratio.

$$E_{offchip\,in\,act\,mem} = \frac{MemCh \times P_{DRAM}}{Freq_{compute}} \times Cycles_{offchip\,in\,act} \qquad (7)$$

$$E_{offchip\,wgt\,mem} = \frac{MemCh \times P_{DRAM}}{Freq_{compute}} \times (Cycles_{offchip\,wgt} + Cycles_{offchip\,wgt\,grad}) \qquad (8)$$

$$E_{offchip\,out\,act\,mem} = \frac{MemCh \times P_{DRAM}}{Freq_{compute}} \times Cycles_{offchip\,out\,act} \qquad (9)$$

$$E_{onchip\,in\,act\,mem} = Cycles_{onchip\,in\,act\,write} \times P_{onchip\,write} \qquad (10)$$

$$E_{onchip\,wgt\,mem} = Cycles_{onchip\,wgt\,read} \times P_{onchip\,read} \qquad (11)$$

$$E_{onchip\,out\,act\,mem} = Cycles_{onchip\,out\,act\,read} \times P_{onchip\,read} + \\ Cycles_{onchip\,out\,act\,write} \times P_{onchip\,write} \qquad (12)$$

$$E_{decomp} = P_{decomp\,(comp\,ratio)} \times \frac{Cycles_{comp\,to\,decomp}}{Freq_{compute}} \qquad (13)$$

$$E_{comp} = P_{comp\,(comp\,ratio)} \times \frac{Cycles_{decomp\,to\,comp}}{Freq_{compute}} \qquad (14)$$

$$E_{decomp\,act} = E_{decomp\,act(comp\,ratio)} \qquad (15)$$

$$E_{decomp\,wgt} = E_{decomp\,wgt(comp\,ratio)} \qquad (16)$$

$$E_{comp\,act} = E_{comp\,act(comp\,ratio)} \qquad (17)$$

| Symbol | Definition |
|---|---|
| $E_{compute\,fwd}$ | Energy consumption of the compute module for the entirety of the computations in the forward pass |
| $E_{compute\,bck}$ | Energy consumption of the compute module for the entirety of the computations in the backward pass |
| $E_{offchip\,in\,act\,mem}$ | Energy consumption of the offchip memory transfers for the network input activations |
| $E_{offchip\,wgt\,mem}$ | Energy consumption of the offchip memory transfers for the network weights |
| $E_{offchip\,out\,act\,mem}$ | Energy consumption of the offchip memory transfers for the network output activations |
| $E_{onchip\,in\,act\,mem}$ | Energy consumption of the onchip memory transfers for the network input activations |
| $E_{onchip\,wgt\,mem}$ | Energy consumption of the onchip memory transfers for the network weights |
| $E_{onchip\,out\,act\,mem}$ | Energy consumption of the onchip memory transfers for the network output activations |
| $E_{read\,ops\,mem}$ | Energy consumption of loading operations from memory |
| $E_{decomp\,act}$ | Energy consumption of decompressing activations in the decompressor |
| $E_{decomp\,wgt}$ | Energy consumption of decompressing weights in the decompressor |
| $E_{comp\,act}$ | Energy consumption of compressing activations in the compressor |
| $P_{decomp\,(comp\,ratio)}$ | Power consumption by the decompressor when loading data from offchip memory at a specific compression ratio (see Table 4) |
| $P_{comp\,(comp\,ratio)}$ | Power consumption by the compressor when writing data to offchip memory at a specific compression ratio (see Table 4) |
| $MemCh$ | Number of available memory channels |
| $P_{DRAM}$ | Power consumption of offchip DRAM |
| $Freq_{compute}$ | Clock frequency of the hardware accelerator |
| $Cycles_{offchip\,in\,act}$ | Compute cycles taken to read input activations from offchip memory |
| $Cycles_{offchip\,wgt}$ | Compute cycles taken to read weights from offchip memory |
| $Cycles_{offchip\,wgt\,grad}$ | Compute cycles taken to read weight gradients from offchip memory |
| $Cycles_{offchip\,out\,act}$ | Compute cycles taken to read output activations from offchip memory |
| $Cycles_{onchip\,in\,act\,write}$ | Compute cycles taken to read input activations from onchip memory |
| $Cycles_{onchip\,wgt\,read}$ | Compute cycles taken to read weights from onchip memory |
| $Cycles_{onchip\,out\,act\,read}$ | Compute cycles taken to read output activations from onchip memory |
| $Cycles_{onchip\,out\,act\,write}$ | Compute cycles taken to write output activations to onchip memory |
| $P_{onchip\,write}$ | Power consumption of a word write to onchip memory |
| $P_{onchip\,read}$ | Power consumption of a word read from onchip memory |
| $Cycles_{comp\,to\,decomp}$ | Compute cycles taken to decompress compressed data |
| $Cycles_{decomp\,to\,comp}$ | Compute cycles taken to compress data |

Table 5: Symbols definition table

## C  QUANTUM MANTISSA – MORE DATA

In this section, we expand the discussion of the effects of *Quantum Mantissa* on the training process. We first analyze the effects in detail on ImageNet, and then follow up with other tasks.

### C.1  IMAGENET

We show the effects of changing the regularization parameter, multiple training runs with the same proposed regularization parameter ($\gamma = 0.1$), stopping early, as well as the choice of whether to simply chop off the removed bits or round them to the least significant remaining bit. These effects are summarized in Table 6.

### C.1.1 EFFECT OF THE γ HYPERPARAMETER

In order to demonstrate the effect of the newly introduced hyperparameter $\gamma$, we run three full training runs with varying $\gamma \in \{0.01, 0.1, 1.0\}$. All other hyperparameters are the same. Figure 7 shows that all runs are able to follow the baseline accuracy and provide significant memory footprint reductions. The cumulative training memory footprint and final validation accuracies are reported in Table 6. The most aggressive version ($\gamma = 1.0$) achieves $40\times$ memory footprint reduction and reduces most layers activations to 0 bits (only exponent bits remain). However, there is a noticeable accuracy degradation of $1.7\%$. The less aggressive version ($\gamma = 0.01$) greatly reduces memory albeit, as expected, to a lesser degree compared to $\gamma = 1.0$. Our selected value ($\gamma = 0.1$) avoids both of these pitfalls, it matches the baseline accuracy and provides $9\times$ compression ratio.

The $\gamma$ parameter is another hyperparameter that needs tuning. However, in our experience a good one is easy to guess since $\gamma = 0.1$ seems to work really well across different models and tasks. Its broad applicability and usefulness is confirmed by the discussion above, other models in Section 3.1 and Section C.2. In fact, all results presented in the main-body of the paper (Section 3.1) use $\gamma = 0.1$.

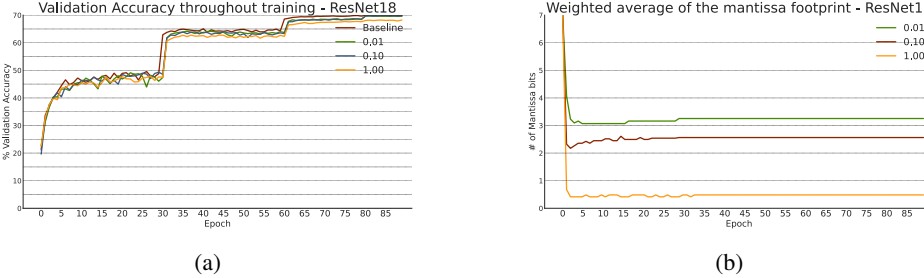

(a)             (b)

Figure 7: *Quantum Mantissa* $\gamma$ effects on ResNet18/ImageNet: (a) TOP-1 validation accuracy during training. (b) Weighted average mantissa length during training.

### C.1.2 CONSISTENCY ACROSS TRAINING RUNS

To demonstrate the consistency of results across different runs we trained ResNet18 3 times with the same hyperparameters, specifically $\gamma = 0.1$. Figure 8 shows that all runs consistently follow the baseline accuracy and exhibit consistent memory footprint reduction. Ultimately, the final accuracy is consistent (standard deviation of $0.13\%$) and exactly matches the baseline with the mean of $69.94\%$ across all the *Quantum Mantissa* runs (Table 6).

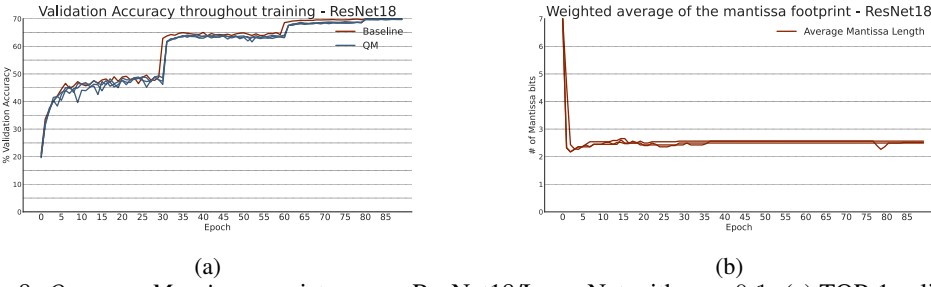

(a)             (b)

Figure 8: *Quantum Mantissa* consistency on ResNet18/ImageNet with $\gamma = 0.1$: (a) TOP-1 validation accuracy during training. (b) Weighted average mantissa length during training.

### C.1.3 CHOICE OF THE BIT REMOVAL METHOD

In addition, we discuss the way of removing the bits selected for removal. Accuracy and memory footprint can be improved by rounding. The rounding version, denoted by R, rounds the least significant remaining bit instead of just ignoring the removed bits. This modification allows *Quantum Mantissa* to be even more aggressive by reducing the footprint by an additional $35\%$ to achieve a footprint reduction of $93\%$, without noticeable loss of accuracy (Table 6). Since the rounded version provides the best accuracy and footprint trade-off, we present it as the *Quantum Mantissa* approach in the main body of the paper.

Table 6: Accuracy and relative mantissa footprint of different *Quantum Mantissa* runs with ResNet18 on Imagenet. Runs labeled with R use rounding instead of chopping. Runs labeled with RES use rounding instead of chopping and stop learning, round up and fix the mantissa length at epoch 15. Experiments with multiple runs show mean and standard deviation of obtained results.

| Version | TOP-1 Validation Accuracy | TOP-5 Validation Accuracy | Relative Mantissa Footprint vs FP32 |
|---|---|---|---|
| Baseline | 69.94 | 89.34 | 100% |
| $\gamma = 0.01$ | 69.76 | 89.21 | 13.9% |
| $\gamma = 0.10$ | $69.94 \pm 0.13$ | $88.30 \pm 0.07$ | $11.1\% \pm 0.1\%$ |
| $\gamma = 1.00$ | 68.25 | 88.40 | 2.5% |
| $\gamma = 0.10$ R | $69.73 \pm 0.26$ | $89.17 \pm 0.1$ | $7.0\% \pm 0.7\%$ |
| $\gamma = 0.10$ RES | 69.50 | 89.05 | 7.9% |

### C.1.4 EARLY STOPPING OF *Quantum Mantissa*

Finally, we present the rounding early stop version, denoted by RES. This version stops the mantissa length learning at epoch 15, rounds them all up, and keeps them constant throughout the last 75 epochs. As a result, the overhead (in our case negligible) is constrained only to the first 15 epochs, with a small drop in accuracy and memory footprint reduction (Table 6). This experiment shows that ending *Quantum Mantissa* early is a viable strategy in cases where overhead is an issue.

### C.2 OTHER TASKS

In this section, we further analyze the effects of *Quantum Mantissa* on tasks such as recommendation systems and natural language processing as well as different network architectures such as transformers. All models we discuss in this section are extremely weight heavy, as opposed to activation heavy ImageNet CNNs. As a result, the weights transfer on and off chip (sometimes repeatedly transferred back and forth in chunks) will be the costliest operation. In this case, reducing weights is much more important. *Quantum Mantissa* excels here due to its ability to zero in on the costliest tensor during training through its additional loss that targets minimum footprint. This is clearly shown in the following section and tables 7 and 8.

### C.2.1 RECOMMENDATION

We present DLRM on the Kaggle Criteo Dataset in Table 7. Since DLRM is trained only on one epoch, we tried *Quantum Mantissa* with stopping the mantissa length learning, rounding and fixing at about 10% and 40% iterations. Both work well. We try $\gamma$ of 0.1 and 1.0. All versions match the baseline and excel at reducing mantissa footprint. Mantissa compression rates are about $10\times$. *Quantum Mantissa* practically removes all mantissa bits from many layers, only leaving the sign and the exponent.

Since DLRM is trained only for one epoch, it does not allow for enough iterations to show the full potential of *Quantum Mantissa*. Consequently, we report the last iteration footprint as well. All iterations beyond the first epoch will exhibit this smallest footprint.

Table 7: Accuracy and relative mantissa footprint of *Quantum Mantissa* runs with DLRM on the Kaggle Criteo Dataset. *Quantum Mantissa* is disabled and mantissa lengths are fixed at the indicated iteration

| Version | TOP-1 Validation Accuracy | Relative Mantissa Footprint vs FP32 | Relative Mantissa Footprint vs FP32 — last iteration |
|---|---|---|---|
| Baseline | 79.42 | 100% | 100% |
| $\gamma = 0.10$, Stop at iteration 30k/300k | 79.50 | 12.2% | 11.5% |
| $\gamma = 1.00$, Stop at iteration 30k/300k | 79.46 | 10.8% | 10.7% |
| $\gamma = 0.10$, Stop at iteration 120k/300k | 79.45 | 11.8% | 11.1% |
| $\gamma = 1.00$, Stop at iteration 120k/300k | 79.43 | 10.7% | 9.6% |

### C.2.2 NATURAL LANGUAGE PROCESSING

Finally, to show applicability of *Quantum Mantissa* on natural language processing we present finetuning of BERT on the MRCP dataset and GPT–2 on the Wikitext-2 dataset in Table 8. We finetuned BERT and GPT–2 for 5 and 3 epochs, respectively. Both models are trained with *Quantum*

*Mantissa* enabled for the first epoch and disabling it, rounding up and fixing the mantissa lengths for the rest. Again, we run both with $\gamma = 0.1$ and $\gamma = 1.0$, and get similar results. We repeated these experiments three times. The general $\gamma = 0.1$ works great all round, matching the baseline accuracy and achieving $8 - 11\times$ mantissa compression, while $\gamma = 1.0$ achieves even better compression ratios at a small cost to performance. At $\gamma = 1.0$ *Quantum Mantissa* manages to find many layers that do not need any mantissa bits.

Similarly to DLRM training, finetuning does not allow for enough iterations to show the full potential of *Quantum Mantissa*. Consequently, we report the last iteration footprint as well. All iterations beyond the predetermined epoch length will exhibit this smallest footprint. This last iteration is a very good estimate of mantissa reduction during a full training run.

Table 8: Accuracy/Perplexity and relative mantissa footprint of different *Quantum Mantissa* runs with BERT and GPT2 finetuning. Values are reported as mean standard deviation across multiple runs. For Perplexity, lower is better.

| Version | Metric | Score | Relative Mantissa Footprint vs FP32 | Relative Mantissa Footprint vs FP32 — last iteration |
|---|---|---|---|---|
| **BERT Baseline** | Accuracy | 84.56 | 100% | 100% |
| **BERT** $\gamma = 0.1$ | Accuracy | $84.31 \pm 0.00$ | $9.34\% \pm 0.01\%$ | $7.82\% \pm 0.01\%$ |
| **BERT** $\gamma = 1.0$ | Accuracy | $86.68 \pm 0.28$ | $3.43\% \pm 0.01\%$ | $1.25\% \pm 0.01\%$ |
| **GPT–2 Baseline** | Perplexity | 20.95 | 100% | 100% |
| **GPT–2** $\gamma = 0.1$ | Perplexity | $20.96 \pm 0.00$ | $12.3\% \pm 0.00\%$ | $11.2\% \pm 0.00\%$ |
| **GPT–2** $\gamma = 1.0$ | Perplexity | $21.42 \pm 0.00$ | $2.68\% \pm 0.00\%$ | $2.0\% \pm 0.00\%$ |

## D  BITCHOP – MORE DATA

This section analyzes in additional detail the effects that *BitChop* has on the training process. We study: a) the effect of the exponential decay factor ($\alpha$) used in the moving average, b) the variation in accuracy across different runs, and c) the effect of the threshold for the change in the moving average.

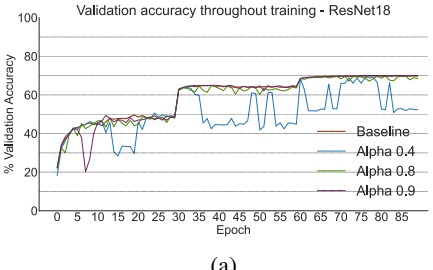
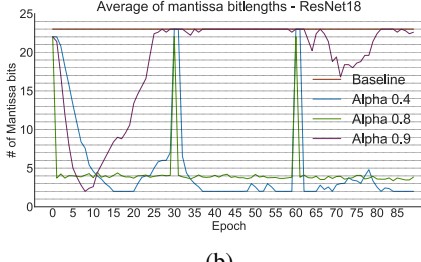

(a)                                            (b)

Figure 9: *BitChop* $\alpha$ effects on ResNet18/ImageNet: (a) TOP-1 validation accuracy during training. (b) Average mantissa bitlengths during training.

Table 9: Accuracy and relative mantissa footprint of different *BitChop* runs with ResNet18 on Imagenet. Experiments with multiple runs show mean and standard deviation of obtained results.

| Version | TOP-1 Validation Accuracy | TOP-5 Validation Accuracy | Relative Mantissa Footprint vs FP32 |
|---|---|---|---|
| Baseline | 69.94 | 89.34 | 100% |
| $\alpha = 0.4$ | 66.27 (inconsistent) | 86.95 (inconsistent) | 11.2% |
| $\alpha = 0.8$ | $69.93 \pm 0.15$ | $88.64 \pm 0.09$ | $23.7\% \pm 2\%$ |
| $\alpha = 0.9$ | 70.07 | 89.37 | 82.6% |

**Exponential decay factor $\alpha$:** We run three full training runs with varying $\alpha \in \{0.4, 0.8, 0.9\}$. All other hyperparameters remain the same. The lower the $\alpha$ the more influence past changes to the loss have on BitChop's decisions and the more resistant it becomes in changing course. Conversely, the higher the $alpha$ the more influence recent changes to the loss are, and the more re-active BitChop becomes. This is reflected in the changes in the validation accuracy and the mantissa bitlengths as

seen in Figure 9. The results show that higher $\alpha$ values better track baseline accuracy as BitChop quickly tries to recover from any apparent increase in loss. However, too high an $\alpha$ value coupled with the natural variations in the loss across different batches prevents BitChop from trimming mantissas significantly as can be seen for $\alpha = 0.9$. Lower $\alpha$ values such as 0.4 produce erratic behaviour in validation accuracy. In this case, the heuristic is looking too far back into previous loss values and becomes slow to change course in adjusting bitlengths leading to trimming them too aggressively. A mid-high $\alpha$ value of $0.8$ balances reactiveness to changes in the loss and bitlength reduction aggressiveness. The cumulative training memory footprint and final validation accuracies are reported in Table 9.

In summary, BitChop works best with an $\alpha$ value of 0.8, as it achieves low bitlengths while being consistent and matching the baseline validation accuracy. Values higher than 0.8 result in almost no bitlength compression. As such, $\alpha$ values between 0.6 and 0.8 were found to strike a good balance between high memory compression and consistent and converging validation accuracy.

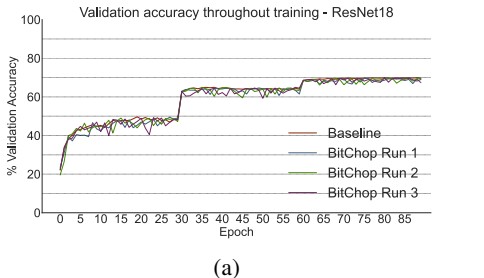 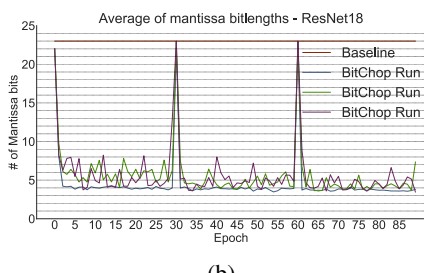

|(a)|(b)|

Figure 10: *BitChop* consistency on ResNet18/ImageNet with $\alpha = 0.8$: (a) TOP-1 validation accuracy during training. (b) Average mantissa bitlengths during training

**Variation Accuracy Across Runs:** Figure 10 demonstrates the robustness of BitChop by reporting how validation accuracy varies across multiple runs (all using the same hyperparameters). All runs of BitChop converged into baseline accuracy with a standard deviation of $\pm 0.15\%$.

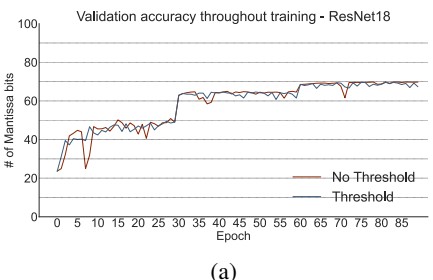 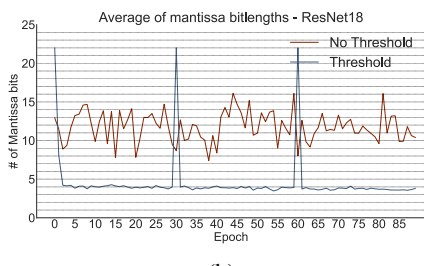

|(a)|(b)|

Figure 11: *BitChop* moving average heuristic with and without threshold on ResNet18/ImageNet with $\alpha = 0.8$: (a) TOP-1 validation accuracy during training. (b) Average mantissa bitlengths during training.

**The Effect of Having a Threshold:** Figure 11 shows the effect of using the threshold in *BitChop*'s moving average heuristic. This shows that the threshold allows *BitChop* to achieve much lower bitlengths while achieving much lower swings in bitlength and accuracy throughout the process. Without the threshold, BitChop becomes overly reactive to minute changes producing an overall erratic behavior. Those swings in turn produce worse overall bitlengths and less memory compression.

**Takeaways:** There are several key takeaways from this ablation analysis:

- Exponential decay factor ($\alpha$) values lower than 0.6 take into account loss values that are no longer useful to tune the bitlength, resulting in a method that is too resistant in changing course and as such reduces bitlengths too aggressively, hurting accuracy. Values higher than 0.8 overemphasize recent loss behavior and as such the heuristic becomes too reactive to minor increases in loss and does not achieve significant memory compression.

- The use of a threshold enables *BitChop* to be resistant to minor changes in the loss and as a result enables it to trim bitlengths more effectively than a heuristic without it, while maintaining baseline accuracy. Otherwise, *BitChop* exhibits unnecessary big swings in bitlength which as a result don't allow it to converge into the low bitlengths seen with the dynamic threshold.

- Crucially, *BitChop* is robust as it keeps validation accuracy convergence and bitlengths consistent over different training runs with the same hyperparameters.

# E  MODEL HYPERPARAMETERS AND DATASETS

In this section we summarize the hyperparamters we used for training and finetuning experiments.

## E.1  RESNET18 AND RESNET50

- Epochs: 90
- Learning Rate: 0.1, 0.01 and 0.0001 at epochs 0, 30 and 60, respectively
- Batch size: 256 and 48 for ResNet18 and ResNet50, respectively
- Momentum: 0.9
- Weight Decay: 1e-4
- Dataset: ImageNet

## E.2  MOBILENET V3 SMALL

- Epochs: 150
- Learning Rate: Cosine Annealing Schedule with starting Learning rate of 0.05
- Batch size: 256
- Momentum: 0.9
- Weight Decay: 1e-4
- Dataset: ImageNet

## E.3  BERT

We use the default hyperparameters provided by Hugging Face for the MRCP dataset.

- Epochs: 5
- Batch size: 32
- Learning Rate: 2e-5
- Dataset: MRCP

## E.4  GPT–2

We use the default hyperparameters provided by Hugging Face for the Wikitext 2 dataset.

- Epochs: 3
- Batch size: 8
- Dataset: Wikitext 2

## E.5  DLRM

We use the default hyperparameters provided by Facebook for the Kaggle Criteo dataset.

- Epochs: 1
- Learning Rate: 0.1
- Mini batch size: 128
- Bottom architecture: 13-512-256-64-16
- Top architecture: 512-256-1
- Dataset: Kaggle Criteo

