# OpenReview forum: "Schrödinger's FP: Training Neural Networks with Dynamic Floating-Point Containers"
_ICLR.cc/2023/Conference — Submitted to ICLR 2023_

### Official Review · Reviewer_hmeb · 2022-10-27

**Confidence:** 2
**Correctness:** 4
**Technical Novelty And Significance:** 3
**Empirical Novelty And Significance:** 3
**Recommendation:** 5

**Clarity, Quality, Novelty And Reproducibility:**

- The paper was not very clearly written, there were several sections that were difficult to parse.
- This is a medium-quality paper with lots of references to contemporary work in the area. The authors do a great job situating their work within the rest of the research in the community. However, the paper still needs a little polishing.
- To my knowledge this approach is novel.
- I have not tried to reproduce this work.

# Minor Points
- Is equation (3) supposed to be $m_i$ instead of $n_i$ and what is the sum over?
- In Figure 2b the x-axis is clipped.
- Is Fig 4a the distribution or the CDF?

**Strength And Weaknesses:**

# Strengths
- The paper addresses an interesting problem that seems to suggest a significant opportunity for making training large-scale models more efficient and performant.
- The results are very compelling.
- The techniques seem simple to implement.
- The authors have done a great job providing references and situating their work within the rest of the contemporary research in the community.

# Weaknesses
- The paper was not very polished and difficult to read.
- The description for quantization could be improved. I still don't understand how it works. Perhaps an example would help.
- What does it mean to fix the bitlengths for some training time to fine-tune the network to this state?
- Error bars and standard deviations are not calculated.
- I would like to see a fair comparison of Quantum Mantissa used on a single quantization level for the entire network with BitChop.
- I would like to see more ablations on the experiments tried for the decisions of BitChop.
- Fig 2c suggests that each network layer should have a different quantization level, but it seems like BtChop only needs a single quantization level for the entire network. The reason for this is not clearly explained.
- For Gecko, since the distribution is already available why not use an optimal value-based lossless encoding like a Huffman encoding scheme

**Summary Of The Paper:**

The paper presents two novel methods for performing dynamic quantization of the activations and weights used during training. That is the authors suggest two ways to dynamically calculate the number of bits used to compress the activations and weights during training. The first method, Quantum Mantissa, adds an additional term to the loss, so the model is able to learn the required quantization for each layer during training. The authors show that this approach can reduce the footprint significantly without sacrificing validation accuracy while incurring only modest overheads. The second method, BitChop, is a less intrusive method that greedily attempts to decrease the network-wide bitlength while the loss keeps decreasing and increases the bitlength when the loss increases. They show that BitChop also decreases the footprint significantly incurring even smaller overheads. The authors also introduce a lossless delta encoding scheme to encode the exponent bits by taking advantage of the non-uniformity of the exponent values. They call this approach Gecko. By combining the mantissa encoding schemes and Gecko the authors evaluate the entire model, which they call Schrodinger FP, on ResNet18 and MibileNet V3 Small on Imagenet. They show that Schrodinger FP has similar performance to the full model but with a significantly lower footprint that leads to a more performant and energy-efficient training scheme.

**Summary Of The Review:**

While the paper tackles a very interesting problem and has very compelling results, I don't think the paper is ready for publication in its current form. The prose needs some work, and there need to be more experiments as well as standard error measurements before the paper is ready for publication.

---

> ### Author Response · Authors · 2022-11-08
> **Reply to Reviewer hmeb**
>
> Thank you for your time to review and discuss our paper! We summarize and respond to your comments below. Please let us know if anything is missing/unclear, if we need to expand our answers, or if you have additional questions/concerns.
>
> *Weaknesses:*
> 1) *Not polished & difficult to read.*
>
>     This work is very dense and interdisciplinary (ML, systems, hardware architecture, and semiconductors). We will try to improve readability. Given the nature of the subject, it may require a few passes to be best appreciated.
>
> 2) *Improve quantization description. Example?*
>
>     Exponents are compressed losslessly. Least significant bits of the mantissa are trimmed. BitChop (BC) and Quantum Mantissa (QM) determine how many bits are needed. BC does this heuristically by observing the loss. QM parametrizes the bitlength, expands it to the continuous domain and directly penalizes it in the loss, and then uses gradient descent to learn it with the other parameters.
>
>     An example. Bfloat16 has 7 bits. QM or BC may decide we only need 3 bits, so we discard the least significant 4. All on-chip operations use padded values and are done in Bfloat16. The short tensor goes off-chip. When it’s back on-chip, the 3 bit mantissa is padded to restore the Bfloat16 datatype.
>
> 3) *Fix the bitlengths near the end?*
>
>     We defined a stochastic datatype for accelerating training. When deployed, a deterministic datatype is preferable, so we disable QM in the last 10 epochs and treat it as a fixed datatype (# of bits determined by QM). No extra epochs. Disabling QM removes its small overhead and retains the benefits.
>
> 4) *Errors and STDs?*
>
>     We haven’t gathered these stats. We will try to show with more data how bitlength and accuracy vary.
> * On ResNet18, over 5 runs with BC, accuracy varied up to 0.15% (STD=0.04); average bitlength varied up to 0.5 (STD=0.21).
> * We will try to update the stats for QM on ResNet18.
>
> 5) *QM vs BC, network level datatype?*
>
>     We can look at Fig. 2c. The largest bitlength QM found is 4 for first and last layers. A per-network approach needs to support the worst case, so QM would likely select 4 bits everywhere. On average, BC found a bitlength of 4.5. While it is 12% more, BC does remarkably well for its simplicity.
>
> 6) *BC ablations?*
>
>     We developed several heuristics for BC; we only reported the best heuristic and its best hyperparameters due to space constraints. We will add an in-depth ablation discussion in the appendix.
>
>     Alphas (exponential decay factor) lower than 0.6 provided a bigger footprint as they gave more weight to older loss values, resulting in BC changing bitlength at a slower pace and achieving larger bitlengths. Alphas higher than 0.9 make newer loss values have more impact on the moving average, creating a more erratic and aggressive behavior causing an accuracy drop. Alphas between 0.7 and 0.8 work best for all tested networks, so we showed results from 0.8. A static threshold was ineffective due to the big variations that can happen in the loss across every batch or training step, so we implemented a dynamic threshold based on relative error.
>
> 7) *Coarse vs Fine grained approach? BC is okay with per-network but Fig 2c suggests per-layer is better?*
>
>     QM’s per-layer approach is more space efficient. Finer grained approach leads to more footprint reduction potential. The larger the group of values sharing a datatype, the more values share the largest datatype since it needs to support the worst case. BC is unable to find mantissa lengths on a fine granularity due to its simplicity, but it still does very well. QM, on the other hand, is able to target mantissa lengths at arbitrary granularities and we show it on a per-layer basis as it works well and is simple to exploit. But QM can go finer.
>
> 8) *We know the distribution, why not use an optimal value-based lossless encoding like a Huffman encoding?*
>
>     We use distribution only as a motivation. Distribution is not available during training, the values themselves are generated but the volume is so large that calculating the distribution on the fly is not feasible. Distributions will change during training and from run to run. If we already know the distribution, we already trained the network, defeating the purpose of accelerating training.
>
>     On a side note, Huffman loses its benefits as the ratio of values strays away from the powers of 2 and is therefore not a good fit. Arithmetic encoding would fix it with more complexity, but the problem of not knowing the distribution remains.
>
> *Minor Points*
>
> 1) *Eq.3, mi instead of ni? Sum?*
>
>     No. The ni represents the per-group mantissa length that we learn. The m in Eq.1 represents the max number of mantissa bits. For FP32 it is 23; for Bfloat16 it is 7. The sum is over i (all the different groups). Sorry about the confusion, we will clarify it in the draft.
>
> 2) *Fig.2b x-axis*
>
>     Thank you! We will rectify it.
>
> 3) *Fig 4a distribution or CDF?*
>
>     CDF. We will clarify it.

---

> ### Author Response · Authors · 2022-11-19
> **Post Revision Comment**
>
> Hi Reviewer hmeb,
>
> We have added a revision with more data and a comment stating the changes. In particular, we try to address your concern regarding variability of results in Appendices C (Quantum Mantissa) and D (BitChop). In addition, Appendix D contains more info on different experiments we conducted with BitChop.

---

### Official Review · Reviewer_5HSw · 2022-10-29

**Confidence:** 4
**Correctness:** 3
**Technical Novelty And Significance:** 2
**Empirical Novelty And Significance:** Not applicable
**Recommendation:** 3

**Clarity, Quality, Novelty And Reproducibility:**

Although combined together, the proposed three quantization techniques seem to be separately proposed with incremental ideas, limiting the novelty of this paper. Also, the hardware benefits are hard to verify given the limited discussion about the proposed hardware architecture and experiment settings.

**Strength And Weaknesses:**

(Strengths)
- The concept of customizing mantissa and exponent bits would be critical for improving the efficiency of challenging DNN training.

(Weaknesses)
- Although the proposed Quantum-Mantissa method is based on parameterization of bit-width for the mantissa, there is no in-depth analysis of how well this parameter is optimized. Also, there is no discussion on the impact of these techniques on the training convergence

- The proposed methods are mainly applied to forward-pass computation, and thus the scope of computational savings is limited. Also, the authors presented the energy savings based on their hardware implementation, but it is hard to thoroughly verify the claimed advantages given the limited information provided by the discussion.

- The evaluation is very limited; only covers two CNNs with typical types. So, it is hard to generalize the proposed methods.

**Summary Of The Paper:**

This paper proposed reduced-precision quantization methods for efficient deep neural network training. One of the methods mainly proposed in this paper parameterized the number of mantissa bits so that the number of mantissa bits can be automatically adjusted during training (Quantum-Mantissa). The authors also proposed another (more computationally simple yet less effective) mantissa bit decision algorithm based on the difference in the loss (BitChop). Lastly, the authors proposed a lossless exponent bit encoding (Gecko). The authors claim that the proposed method can reduce memory transfer while maintaining original model accuracy after training.

**Summary Of The Review:**

The authors proposed a set of bit-reduction methods for floating point format, but without sufficient theoretical analysis and empirical justification. It would be highly appreciated if the authors add more in-depth discussion on the algorithmic success of the proposed methods and if they demonstrate this benefit throughout more variety of applications.

---

> ### Author Response · Authors · 2022-11-07
> **Reply to Reviewer 5HSw**
>
> Thank you for your time to review our paper and discuss it! We summarized your questions and concerns below and responded to each. Please let us know if anything is missing or unclear, if we need to expand any of our answers, or if you have any additional questions or concerns.
>
> *Weaknesses:*
>
> 1) *How well is mantissa optimized? Impact on training convergence?*
>
>     We show that for our experiments there is a negligible impact on accuracy...
>
>     We will try to add the experiments considering they do take time to run. However, we do show that the method is robust and effective for a widely used class of models. Given the interdisciplinary nature of the work and the space available, there are several considerations that we had to cover.
>
> 2) *Methods are mainly applied to forward-pass computation, and thus the scope of computational savings is limited.*
>
>     It’s not limited, it’s zilch :)
>
>     We do not touch the computation; we “just” transparently chop off or zero out the N least significant mantissa bits, and reduce the number of bits used for encoding the exponents when we store the results in memory. When the numbers are read back, they are expanded to the original format.
>
>     Training speed and energy are dictated by off-chip memory. Figure 7 in the Gholami et al. paper, Ref [2] mentioned by Reviewer JoBj (#2), clearly shows (on a log scale) the energy costs of different operations. Reducing compute will be negligible unless we also reduce memory. With our methods, we shift the balance to compute. Our method makes it profitable to look at computation reduction methods in future work.
>
> 3) *Hard to thoroughly verify the advantages.*
>
>     This is interdisciplinary work. We use best practice simulation and circuit modeling (down to layout -- post synthesis with commercial tools and technology node).
>
>     Our simulation method is explained in the Appendix B and is standard practice in the computer architecture community for this level of analysis.
>
> 4) *The evaluation is very limited; only covers two CNNs with typical types. How does it generalize?*
>
>     We will try to update the draft with more diverse models. But given the novelty of the work, and the fact that vision models have so many applications, we believe this is enough evidence to demonstrate our method is useful. We do not claim broad applicability, although it's hard to imagine that it won’t apply to different models.
>
> *Limited novelty?*
>
> Please see the general comment on novelty.
>
> **Reference from Reviewer JoBj (#2)**
>
> [2] Gholami et al., "A Survey of Quantization Methods for Efficient Neural Network Inference", 2021

---

> ### Author Response · Authors · 2022-11-19
> **Post Revision Comment**
>
> Hi Reviewer 5HSw,
>
> We have added a revision with more data and a comment stating the changes. In particular, we try to address your concern regarding impact on training in Appendicies C (Quantum Mantissa) and D (BitChop), and concern regarding the limited model diversity by exapnding it in Section 3.

---

### Official Review · Reviewer_JoBj · 2022-11-03

**Confidence:** 3
**Correctness:** 2
**Technical Novelty And Significance:** 2
**Empirical Novelty And Significance:** 2
**Recommendation:** 5

**Clarity, Quality, Novelty And Reproducibility:**

The paper's clarity and quality can be significantly improved as mentioned above. The novelty is also marginal. The authors did not open-source their code yet for reproducibility but that did not affect my rating.

**Strength And Weaknesses:**

Strengths:

1. The paper is generally well-motivated and the problem chosen is timely, given the adverse environmental impact of training of large-scale ML models.

Weaknesses:

1. The paper is hard to read at times. For example, in the abstract, the authors pose Schrodinger's FP as one of the contributions, but it is only described in the supplementary materials. It is very unclear to me what is the encoding/decoding scheme, how does the bit-length reduction results in the memory footprint reduction, etc.
2. The paper's contributions look incremental. For example, the stochastic quantization scheme employed for Quantum mantissa seems similar to a wide range of works [1-2]. The loss function to learn the optimal bit-length is not new either [3]. In fact, there are also works that progressively reduce the bit-width of the network during training [4].
3. The authors did not compare their results with integer (fixed point) quantized neural networks which can even further reduce the memory footprint. I am curious what might be the accuracy drop compared to the proposed quantized FP16 approach.

[1] Dong et al., "Learning Accurate Low-Bit Deep Neural Networks with Stochastic Quantization", BMVC 2017

[2] Gholami et al., "A Survey of Quantization Methods for Efficient Neural Network Inference", 2021

[3] Yang et al., "BSQ: Exploring Bit-Level Sparsity for Mixed-Precision Neural Network Quantization", ICLR 2021

[4] Datta et al., "ACE-SNN: Algorithm-Hardware Co-design of Energy-Efficient & Low-Latency Deep Spiking Neural Networks for 3D Image Recognition", Frontiers in Neuroscience 2022

**Summary Of The Paper:**

This paper dynamically adjusts the size and format of the floating point activations and weights using gradient-based approaches to improve the training energy efficiency and execution time of ML models. It also implements encoders and decoders that guided by these approaches reduce the off-chip memory accesses, further boosting the energy efficiency.

**Summary Of The Review:**

See weaknesses above. I am giving a score of 3, but I can reconsider my score based on the authors' rebuttal and discussions with the other reviewers.

---

> ### Author Response · Authors · 2022-11-07
> **Reply to Reviewer JoBj**
>
> Thank you for your time to review and discuss our paper! We summarized and responded to your comments below. Please let us know if anything is missing or unclear, if we need to expand any of our answers, or if you have any additional questions or concerns.
>
> *Weaknesses:*
>
> 1) *Hard to read. Schrodinger's FP? Encoders/Decoders? How to reduce footprint?*
>
>     "Scrodinger’s FP" is the name for our family of methods that includes Quantum Mantissa, BitChop and Gecko. The footprint reduction is directly from using fewer bits per value. Our encoders and decoders detailed in the appendix show how clever packing of values in memory can exploit the full potential of the new adaptable datatype. The memory layout and encoders/decoders are related to Proteus (fixed-point, per-layer precision) [5], Shapeshifter (fixed-point, per-group precision, similar to our exponent encoding) [6] and Samsung’s NPU [7].
>
> 2) *Incremental? Stochastic quantization [1-2]? Loss function for optimal bitlength [3]? Reduction of bitwidths during training [4]?*
>
> * No other work learns and adapts the training datatype during training (see the general comment above).
> * [1] targets achieving the best accuracy during **inference** with a **pre-selected** datatype. It stochastically chooses values that will be quantized. It still uses floating-point for training. While [1] depends on someone else telling them what datatype will work, we **determine** which datatype will achieve baseline accuracy while minimizing the **training** cost. [1] is great in itself, but it is irrelevant in our discussion.
> * [2] is a great summary of quantization. Thank you for pointing it out, as it makes our case much stronger! We will touch on the most relevant sections for our discussion:
>
>     * Section 3.H talks about stochastic quantization. They exclusively discuss stochastic rounding of the **values, not the datatype**. Their datatype is **fixed** and **pre-determined**. Again, our work **determines** the datatype.
>
>     * Section 4.B describes Mixed Precision training. It’s discussed exclusively for **inference**. The methods boil down to RL, NAS, heuristic-based techniques, etc. As mentioned in our introduction, none of these are applicable to **training** because their inherent overheads overshadow any benefits of smaller datatypes.
>
>     * The last paragraph of Section 6 describes previous quantization approaches for efficient training. It shows only broad, pre-determined datatypes like half-precision floating point. It also comments on how hard it is to push down further or use integers broadly for training, which is exactly what our work targets.
>
> * [3] cleverly decomposes fixed point weights into terms and then prunes them to reduce the total bitlength in **inference**. Again, their work does nothing for training, and can’t be simply modified to improve training. It just **guesses** a good common datatype for activations, which are the costliest part of training. Furthermore, it can’t be simply modified to support floating point datatypes which are often required for training.
> * We find no similarity of [4] to our work:
>     * It targets SNN acceleration in a very specific task. They themselves claim that it doesn’t translate well nor saves as much when applied to CNNs (Section 4).
>     * More importantly, the what (2 datatype options) and when (which epochs) are both pre-determined manually. Specifically, it trains in full precision for half of the training and then in quantized 6-bit integers in the second half. Again, all datatypes are **pre-selected** and not determined on the fly. This is one of the many approaches that require hand selection of the datatypes which we referenced in our introduction (e.g. from IBM, Intel, NVIDIA, ARM, & EPFL).
>
> 3) *Comparison with fixed-point?*
>
>     We did not compare with fixed point training since it cannot be reliably and broadly applied to training. [2] clearly states this (see above our answer regarding [2]). Our datatypes are even smaller than INT8 on average. As further evidence in favor of small floating point datatypes, we point to the very recent joint exploration of FP8 by NVIDIA, Intel, and AMD [8].
>
> *Open-source code?*
>
>    Our code is bleeding edge. We are cleaning it up and commit to releasing it upon acceptance.
>
> *Clarity?*
>
>    The work is dense and interdisciplinary, requiring some background in ML, systems, hardware architecture, and semiconductor technology. There is no perfect paper, but given the nature of the subject, it may require a few passes to be best appreciated.
>
> **Additional References:**
>
> [5] Judd et al., "Proteus: Exploiting numerical precision variability in deep neural networks", ICS 2016
>
> [6] Lascorz et al., "Shapeshifter: Enabling fine-grain data width adaptation in deep learning", MICRO 2019
>
> [7] Park et al., "9.5 A 6K-MAC Feature-Map-Sparsity-Aware Neural Processing Unit in 5nm Flagship Mobile SoC", ISSCC 2021
>
> [8] Micikevicius et al., “FP8 Formats for Deep Learning”, arxiv:2209.05433

---

> ### Comment · Reviewer_JoBj · 2022-11-21
> **Rating updated accordingly**
>
> Thanks for your rebuttal response! However, I am still not convinced about the value proposition of the approach and the presentation quality.
>
> 1. If "Scrodinger’s FP" is indeed the name for your family of methods, it should be stated explicitly in the paper, for example, something like "we propose Schrodinger's FP that consists of .....". This should not be brought up at the last sentence of the abstract after all the methods have been introduced. Also, from the text, it is hard to understand what is the proposed approach, and what are the related works. I understand this is quite late, but, it would be nice to clearly bifurcate the proposal and the related works in separate sections in your future submission.
>
> 2. I still think the approach is incremental. Why we can not determine the data type in Yang et al. dynamically as the authors do? In my humble opinion, the same argument holds good for Dong et al.
>
> 3. There are works showing the efficacy of fixed point training in complex vision tasks [1-2], and hence, it warrants a thorough comparison of the training cost of the proposed approach with these works. Moreover, I have also not seen any comparison with low-bit floating point training approaches [3] as well in the paper.
>
> 4. It is a bit stretch to say this work covers semiconductor technology. It is unfortunate that even being a researcher in HW-SW co-design for ML, I find this paper is unclear and lacks clarity. Hence, I strongly recommend the authors to work on this aspect in their future submission.
>
> For all these reasons, I don't think this work is ready for an ICLR publication. However, I would like to increase my rating to 5, and encourage the authors to pursue this line of work.
>
> [1] Zhang et al., "Fixed-Point Back-Propagation Training", CVPR 2020
> [2] Yamaguchu et al. "Training Deep Neural Networks in 8-bit Fixed Point with Dynamic Shared Exponent Management", DATE 2021
> [3] Sun et al. "Ultra-Low Precision 4-bit Training of Deep Neural Networks", NeurIPS 2020

---

> > ### Author Response · Authors · 2022-11-23
> > **Response to JoBj (1/3)**
> >
> > Thank you for your willingness to engage in a discussion with us and for upping your score. You raise some interesting points. We are certain that the points below will help you reconsider further.
> >
> > 2) *“Why can't we determine the data type in Yang et al. dynamically? …” or “Dong et al.”*
> >
> >     Yang et al (BSQ) learn the datatype to use for **weights** during inference. The question posed is: Given BSQ, is it possible to **modify it somehow** to learn and use different datatypes for both activations and weights during training? Answering it requires two steps: 1) Come up with “modifications”, and 2) Evaluate and compare with the resulting BSQ++. We believe that the more proper approach is to present our work as it stands as a complete evaluation of a working method. Anyone can then pursue potentially better methods and have them peer-reviewed. This is in our opinion the best way to handle these considerations.
> >
> >     Nevertheless, to help us all appreciate the magnitude of this non-trivial research task, we highlight some of the challenges that are apparent now — more are likely to emerge once this is pursued.
> >
> >     1) **BSQ only targets weights**, rightfully so since a goal of the work is to improve inference for edge devices. There, weights are the problem. On larger devices for training, ImageNet in particular, weights are a negligible cost. Batch sizes are too large to make targeting weights worthwhile. Our Fig. 5 (a) shows the relative footprint of weights and activations.
> >     2) **BSQ relies on the user to pre-select the activation datatype** (8b and 4b) and uses PACT to quantize them. 8b is used for the first and last layers (see description for Table 3, Pg 8). Even if we move past the fact that their training uses floating point, we can make the following observations: In inference, activations only “live” between layers; this is not the case during training where they have to be stashed and retrieved for the backward pass. Committing oneself to a specific datatype is problematic in two ways: 1) it assumes that training will converge, which, depending on the model, might not be the case, and 2) it prevents us from capturing the full potential in reducing storage.
> >     3) **It is unclear whether BSQ can be converted to target activations**. BSQ learns bitlengths in a two-step process. It first chooses a fixed-point weight datatype, sets the individual bits as learnable parameters, and learns them over a couple of epochs in floating point. After that comes the second step, the precision adjustment. This step first reconstitutes the weights into a single uniformly quantized value and removes the least and most significant zero bits. The process then restarts with the newly selected datatype. Can this be applied to activations that will vary wildly from batch to batch? This is an open question without a clear answer, both on **how** and **whether** it will work.
> >     4) **BSQ introduces significantly larger overheads.** See Pg 4, the 2nd paragraph after equation (3) starting with “The proposed bit…”. For a “maximum potential” bitlength of N, BSQ introduces N operations per element per tensor (to perform gradient descent on each parameter). We introduced just one. That is an order of magnitude difference! For CNNs, their (and ours for that matter) method has negligible overheads since the weight tensors, and the corresponding number of extra operations are much smaller than the total number during training. But what happens in the case of weight-heavy models (e.g., transformers, lstms, mlps…)? Overheads now start to balloon.  Scrodinger’s FP can deal with twice as many computations if we reduce the footprint significantly (and we do), but BSQ will have a hard time keeping up with an overhead that is an order of magnitude larger. Again, all of this is hypothetical and serves to demonstrate that this is a research project that has to be pursued and evaluated properly.
> >     5) **BSQ trains everything in full precision.** We could just use the quantized values in both forward and backward passes. If we did that, and if it worked, it would be incremental. As discussed above, modifying BSQ to target training faces challenges that go far deeper.
> >
> >     To, hopefully, conclude the BSQ discussion. Yang et al. present great work on improving CNN inference. They target the weights (the biggest issue of CNN inference), and they do it cheaply because of the properties of CNN training. BSQ, in its original form, does not improve training in any way and will introduce massive overheads in the case of some models (transformers, etc.). Maybe it can be modified to do so, maybe not. Nevertheless, as outlined above, many challenges are apparent.

---

> > > ### Author Response · Authors · 2022-11-23
> > > **Response to JoBj (2/3)**
> > >
> > > 2) continued:
> > >
> > >     **Dong et al. do not determine the datatype during training. They pre-select it.** There is no way around that fact. They only stochastically decide whether to quantize or not. If they had a crystal ball to tell them what datatype to choose, their work could, with some modifications, do great to improve training. Our work provides this crystal ball.
> > >
> > >     **We ask that you reconsider whether rejecting our work on the basis of demonstrably vague hypotheticals is fair.** We vehemently believe that it is not. And worse, it would create an improper precedent.
> > >
> > > 3) *Sun et al., Zhang et al. and Yamaguchu et al.*
> > >
> > >     So far, all quantization work targeting efficient training has the same problem: “To find the answer, you must know the answer”. You must know which datatype will work, and train with it. Your answer will be confirmed only after you finish training. Or work gives that answer! No trials, no errors, just one efficient training run. This applies not only to Dong et al., but to Sun et al., and Yamaguchu et al.
> > >
> > >     **A common comment about Zhang et al. and Yamaguchu et al. (per method comments will follow):** In the end, they both used 8b for activations and weights. We can do a rudimentary comparison with 8b fixed point, by comparing the memory footprint — the biggest contributor to energy and time cost. **Quantum mantissa and Gecko reduce every single model we tested to below 8 bits.** We can achieve these gains because of our granular approach.
> > >
> > >     If fixed point is enough, and it often isn’t, a simple modification of our work could do wonderfully: BitChop- or Quantum Mantissa- like method on fixed point, and Gecko on top for additional lossless compression. Now this would be a great example of incremental work, but nevertheless, one that should be properly evaluated and peer-reviewed. We believe that publishing our work will enable such follow-up work and further improvements. Not publishing it instead hinders such advances.
> > >
> > >     **Zhang et al.** propose an interesting approach to use int8 (and higher if needed) activations and weights during training. We end up using much less on average. They rely on using FP32 to collect statistics, periodically. They achieve nearly identical accuracy to FP32. They evaluate their method on a CPU with vector extensions. CPUs are more forgiving to overheads as they suffer from several other inefficiencies when executing deep learning workloads. They report speedups on a V100 GPU vs FP32 training. We note that the V100 has significantly higher compute bandwidth for the 16b float operations they use. The paper does not clarify whether that is the primary reason for the observed speedup. Furthermore, it does not compare with FP16 training. Presumably, they would fare even worse on a lean ASIC.
> > >
> > >     **Yamaguchu et al.** propose to use Flexpoint with 8b mantissas (i.e., 8b fixed-point plus a shared exponent). Their contribution is adjusting the meaning of the exponent across training passes to best fit the distribution of the tensors. They require specialized hardware support and a heavily modified training method. Specifically, they have to collect additional statistics on the fly. To update weights, they have to convert them to FP32 and then back to 8b Flexpoint. This requires intrusive hardware and software changes throughout. They show Top-1 accuracy only on graphs where the scale is such that even differences around 5% or less are hardly noticeable, making a direct comparison hard. They do report Top-5 in a table. We work with an almost unmodified training method and our HW additions are surgical: between the on-chip compute and the memory hierarchy and controller.
> > >
> > >     **Sun et al.** propose a very short 4b approach that works **sometimes**. It works for shallow and simple networks. It doesn’t work for deep and complex ones. Their solution? Pick layers by hand to use large datatypes (and build hardware that can handle all these different datatypes). They state this themselves in the last two paragraphs of Section 5, pg 8. On ResNet50, they get awful results (error of 2.5%). After careful handcrafting, they still have an error twice as large as ours (1% vs 0.5%). The results are similarly bad for MobileNet and transformers. After the battle, everyone is a general. Once we train the network many times we will know when and where to use the larger datatype. But what if we have a new architecture, task, or dataset? It’s much harder to correctly guess then. Automating this choice is a problem worth solving! We cover this in detail on the second page of our introduction.

---

> > > > ### Author Response · Authors · 2022-11-23
> > > > **Response to JoBj (3/3)**
> > > >
> > > > 4) *Lack of clarity of HW:*
> > > >
> > > >     We hear you, however, we ask that you reconsider whether this is the case: HW techniques have been advancing for decades and in the case of ML hardware, rapidly in the past 10 years. It is hard to keep up with all advances. At the same time, new advances are often bound to build upon recent ones. Our work is no different: it builds on recent advances and the “how” requires an appreciation of those advances. In our case, each of these past designs required several 11-page papers to be described, motivated, and properly evaluated.
> > > >
> > > >     Here’s the gist of what the writing had to communicate to the reader and our strategy for achieving this goal:
> > > >
> > > >     Our HW compressors perform the following functions:
> > > >
> > > >     * Determine how many bits the exponents can be packed in, i.e. find the maximum exponent per group of 8;
> > > >     * Strip as many least significant bits from the mantissa as instructed by Quantum Mantissa or BitChop;
> > > >     * Pack the values tightly and write them to memory;
> > > >     * The HW Decompressors reverse the above.
> > > >
> > > >     The key challenge in the above is how to pack variable bitlength values in the memory row so that we can avoid a large crossbar and shuffling interconnect to pack and unpack the values.
> > > >
> > > >     The challenge at first appears to be hard; however, it has been extensively addressed in past works and we think you will find the key concept to be a very useful tool as this challenge appears in many different contexts for accelerating ML. We provide the reader with a fairly detailed description in the appendix and further give them tools to dive deeper once they are ready to invest the time.
> > > >
> > > >     Specifically, we point to the following two very relevant works that we build upon:
> > > >
> > > >     * The Proteus [5] work from 2016 devotes a whole section to the idea of virtual columns, yet for fixed-point and one precision per layer.
> > > >     * Shapeshifter [6] from 2019 shows how to detect precisions per group and pack them in memory with little cost (the core concept was discussed in Dynamic Stripes in 2017 [9]).
> > > >
> > > >     Both the designs above work with fixed-point. They invest a considerable part of the paper to explain *how* this is done. Samsung’s ISSCC paper [7] also shows how they implement it in their flagship NPU. We cannot provide such a detailed description and at the same time present Quantum Mantissa and BitChop. We adapt these designs to dynamically adjust only the exponent part and accept the mantissa length from Quantum Mantissa and BitChop.
> > > >
> > > >     Since you do work in HW too, maybe, like us, you had to invest extra time to understand how several HW techniques such as out-of-order execution, or load-store queues work under the hood. This is one of these cases where the reader has to invest additional effort to follow up with prior work to appreciate the summary given in the paper.
> > > >
> > > >     Incidentally, the mixed-precision training papers that came up in our discussion do not discuss at all or in sufficient detail how they could pack values of different bitlengths when reading or writing from memory.
> > > >
> > > >     In conclusion, we hope that you will agree that we provide the reader with the necessary tools (summary and references) to appreciate the hardware aspects of the work. We commit to releasing the Verilog for the designs.
> > > >
> > > > 1. *Stating early on that the name refers to a set of methods: something like we "propose Schrodinger's FP that consists of ....".*
> > > >
> > > >     We thought we communicated the idea in the Introduction (Pg 2, 4th paragraph: “Our solution is Schrodinger’s FP, a family of methods…”). Unfortunately, we didn’t manage to get our message through our writing.
> > > >
> > > >     Different people have different styles and we strive to improve. We hear you and we will modify it as you suggest.
> > > >
> > > > **References from earlier:**
> > > >
> > > > [5] Judd et al., "Proteus: Exploiting numerical precision variability in deep neural networks", ICS 2016
> > > >
> > > > [6] Lascorz et al., "Shapeshifter: Enabling fine-grain data width adaptation in deep learning", MICRO 2019
> > > >
> > > > [7] Park et al., "9.5 A 6K-MAC Feature-Map-Sparsity-Aware Neural Processing Unit in 5nm Flagship Mobile SoC", ISSCC 2021
> > > >
> > > > **Additional Reference:**
> > > >
> > > > [9] Lascorz et al., "Dynamic Stripes: Exploiting the Dynamic Precision Requirements of Activation Values in Neural Networks", arxiv 2017

---

### Official Review · Reviewer_pQea · 2022-11-03

**Confidence:** 4
**Clarity, Quality, Novelty And Reproducibility:** The paper is well written and easy to…
**Correctness:** 3
**Technical Novelty And Significance:** 2
**Empirical Novelty And Significance:** 2
**Recommendation:** 5

**Strength And Weaknesses:**

Strengths
1. Scaling model size is the current approach for improving quality, and so the memory footprint of model training is an important problem for training efficiency and costs.
2. Dynamic and adaptive techniques are likely most effective because of high variability of models, hardware, and training phases.
3. Hardware-acceleration is probably the most efficient way to realize these optimization techniques.

Weaknesses
1. I think the evaluation is greatly weakened by only considering image models. In particular, it would have been useful to evaluate the effectiveness on NLP models, which have driven most of the interest in mixed-precision training.
2. The paper does not discuss the handling of overflows/underflows during training. In my experience, these are important issues for mixed-precision training in practice.
3. The opportunity for low-precision representation of training tensors because of their value distribution is not a novel observation and is well-studied. I feel that new insights to the community could be in more efficient exploitation approaches (hardware acceleration is certainly a good direction) and demonstrating generality over a range of model architectures (e.g., transformers) and tasks (e.g., NLP, multi-modal).

**Summary Of The Paper:**

The paper proposes a combination of three hardware-accelerated techniques (Gecko, Quantum Mantissa, and BitChop) for optimizing the memory footprint of model training through low-precision floating point tensors. The approach is inspired by the observation that in some scenarios, the value distribution of training state, such as weights and activations, could be represented with fewer bits without harming the model performance. In particular, the paper proposes both dynamic and adaptive techniques for lossy and lossless quantization of the mantissa and exponents of training tensors.  The evaluation results show memory savings and negligible convergence impact for some image models.

**Summary Of The Review:**

Overall, I feel the paper is lacking in novelty and evaluation.

---

> ### Author Response · Authors · 2022-11-07
> **Reply to Reviewer pQea**
>
> Thank you for your time to review our paper and discuss it! We summarized your questions and concerns below and responded to each. Please let us know if anything is missing or unclear, if we need to expand any of our answers, or if you have any additional questions or concerns.
>
> *Weaknesses:*
> 1) *Evaluation greatly weakened by only considering image models. What about NLP?*
>
>     We will try to update the draft with more diverse models but given the novelty and the interdisciplinary nature of the work, demonstrating it for this important class of models is significant on its own.
>
> 2) *Overflows/underflows?*
>
>     There are none as computations still use the original precision (BF16 or FP32). We just trim mantissas and encode exponents when writing and reading from DRAM (which is ~100x slower and requires ~50x the energy vs on-chip MACs and accesses).
>
>     Indeed, work in mixed-precision training utilizes a variety of hand-selected datatypes and arithmetic. However, our work does not fit this mold: we decouple the choice of the computation datatype and the way the results are stored in memory.
>     Incidentally, we could use lower precision arithmetic, but then we would have to deal with over/underflow and would probably affect accuracy, not to mention that the benefits will be negligible. Figure 7 in the Gholami et al. paper, Ref [2] mentioned by Reviewer JoBj (#2), clearly shows (on a log scale) the energy costs of different operations.
>
> 3) *Low-precision representation for training well-studied? Generality, NLP, transformers?*
>
>     Existing work relies on guessing what datatypes would work and for what part of the computation (the choice is made in advance and is static). That is indeed “well studied” and the experience has been that no method works well for all cases and all prior attempts generalize poorly.
>
>     Ours is a fundamentally different approach which, surprisingly, has received *no* attention. We do not guess or preselect datatypes, we automate them. That is, learning and taking advantage of lower precision and adaptable containers (how many bits to use when storing in memory) on-the-fly during training. Our methods push the limits of what is possible for mantissas. For exponents, they are even better than past methods since on average, we end up using fewer bits than any statically chosen exponent length while being 100% lossless.
>
>     See #1 for generality.
>
> *Summary of the Review:*
>
> 1) *Novelty?*
>
>     Please refer to the "general comment on novelty" above.
>
> **Reference from Reviewer JoBj (#2)**
>
> [2] Gholami et al., "A Survey of Quantization Methods for Efficient Neural Network Inference", 2021

---

> > ### Comment · Reviewer_pQea · 2022-11-22
> > **Review #2**
> >
> > Thanks for your response and clarifications.
> >
> > The inclusion of NLP finetuning results is a step in the right direction, but insufficient to convince that your approach is ready to replace existing mixed-precision approach for NLP pretraining. Although, I feel your approach should more efficiently match the model quality of mixed-precision, because of the extra dynamism, the current draft does not demonstrate this superiority, especially since mixed-precision is scaling to NLP models with trillions of parameters. To be clear, I don't expect pretraining results with multi-billion parameter models, rather I feel that such results could be shown with BERT/GPT2 pretraining and more GLUE fine tuning results.

---

> > > ### Author Response · Authors · 2022-11-24
> > > **Response to Reviewer pQea**
> > >
> > > Thank you too for investing the time to go through our response and revision and for making further suggestions. Please consider the points below:
> > >
> > > 1) We will try the following:
> > >
> > >     * train BERT as suggested (please do consider that this is a long and costly process),
> > >     * include more GLUE results.
> > >
> > >     However, we can gain some insights from our experiments in fine-tuning. The results indicate that we can learn lean datatypes for transformers as well and that we can do it in a rapid manner. Fine-tuning is too short to allow Quantum Mantissa to shine, as shown in Appendix C.2.2, in the last paragraph in particular. There is no reason to believe that the datatype and performance trends will significantly differ when pretraining; if anything, Quantum Mantissa will have more time to squeeze them further.
> > > 2) Mixed-precision methods preselect a datatype before training, while we determine it during training. Both can be useful at the same time. Why view our method as a competitor to training optimizations that use a non-uniform initial selection of datatypes? Our methods **complement and amplify** benefits over mixed-precision training: If one knows in advance dataypes that will work and are smaller than the baseline, why not use them? Why not start from a better baseline? After all, this is what Quantum Mantissa and BitChop do right now at the beginning of every epoch: they start with a mixed-precision state and adjust it further to maintain convergence while reducing footprints. Even if we were to start with a mixed-precision baseline, as our work demonstrates, our methods would work to further trim memory footprints, accelerating training and improving energy efficiency. Recall that we encode exponents using their actual value content and mantissas are often trimmed to 1b or 2b. By demonstrating benefits over FP32 and BFloat16 and specifically by showing that many layers need very short bitlengths, we provide ample evidence for follow-up work to explore the interplay with mixed precision. We also see no reason why our method cannot be adapted to work with fixed-point values (see BitPruning [1]). However, these options have to be properly evaluated and deserve a separate study.
> > > 3) We have demonstrated that our method works for several important models. As such, the work is already useful for at least those application domains. As you suggested, our methods will most likely work for other models as well. While proving a broader applicability would be highly desirable (meaning demonstrating pre-training with transformers), in our opinion, it is not a sufficient reason to reject the work as a whole.
> > >
> > > [1] M. Nikolić, G. B. Hacene, C. Bannon, A. D. Lascorz, M. Courbariaux, Y. Bengio, V. Gripon, and A. Moshovos, “Bitpruning: Learning bitlengths for aggressive and accurate quantization,” 2020

---

> > > > ### Author Response · Authors · 2022-12-02
> > > > **More GLUE results**
> > > >
> > > > Hi Reviewer pQea,
> > > >
> > > > The expanded results for GLUE benchmarks are in the table below. We can no longer update the manuscript on openreview, however, we commit to including these results in the next revision. The results show that our methods consistently match the baseline accuracy and provide robust performance and energy benefits.
> > > >
> > > > |  Task |            Metric            | Hugging Face Baseline | Our Baseline |      QM     |      BC     | QM Speedup | QM Energy Ratio | BC Speedup | BC Energy Ratio |
> > > > |:-----:|:----------------------------:|:---------------------:|:------------:|:-----------:|:-----------:|:----------:|:---------------:|:----------:|:---------------:|
> > > > |  CoLA |     Matthews Correlation     |         56.53         |     55.99    |    57.79    |    55.48    |     2.8    |       1.5       |    4.46    |       1.69      |
> > > > | SST-2 |           Accuracy           |         92.32         |     93.23    |    92.32    |    91.74    |    2.82    |       1.51      |    4.63    |       1.7       |
> > > > |  MRPC |          F1/Accuracy         |      88.85/84.07      |  89.01/84.56 | 88.89/84.31 | 88.91/84.42 |    2.52    |       1.46      |    3.54    |       1.73      |
> > > > | STS-B | Pearson/Spearman Correlation |      88.64/88.48      |  88.92/88.63 | 89.03/88.71 |  88.9/88.53 |    2.82    |       1.51      |    4.33    |       1.68      |
> > > > |  QQP  |          F1/Accuracy         |      87.49/90.71      |  87.46/90.71 | 87.73/90.86 | 87.72/90.77 |    2.83    |       1.51      |    4.59    |       1.7       |
> > > > |  MNLI | Matched acc./Mismatched acc. |      83.91/84.10      |  83.87/84.25 | 83.91/84.29 | 83.96/84.09 |    2.82    |       1.51      |    4.55    |       1.69      |
> > > > |  QNLI |           Accuracy           |         90.66         |     90.54    |    91.18    |    90.41    |    2.82    |       1.51      |    4.62    |       1.7       |
> > > >
> > > >
> > > > In addition, we found an accounting bug in our code that didn’t fully capture all the weight transfers. We report the updated table below for the weight heavy models in the revision. Our methods remain effective energy and execution time performance wise. The accuracy and other measurements remain as they were. The CNNs in both the original draft and the revision are unaffected since weights are so small that the effects are negligible.
> > > >
> > > > |  Network | QM Speedup | QM Energy Ratio | BC Speedup | BC Energy Ratio |
> > > > |:--------:|:----------:|:---------------:|:----------:|:---------------:|
> > > > |   DLRM   |    2.33    |       1.47      |    2.22    |       2.17      |
> > > > |   GPT2   |    2.73    |       1.48      |    3.42    |       1.58      |
> > > > |   BERT   |    2.52    |       1.46      |    3.54    |       1.73      |
> > > >
> > > >
> > > > Sorry for the mixup! Again, we will fix it in the next draft.

---

> ### Author Response · Authors · 2022-11-19
> **Post Revision Comment**
>
> Hi Reviewer pQea,
>
> We have added a revision with more data and a comment stating the changes. In particular, we try to address your concern regarding the limited model diversity and lack of NLP and transformers by expanding it in Section 3.

---

### Author Response · Authors · 2022-11-07
**General Comment on novelty**

Thank you to all reviewers for making the time to read, review and discuss our paper. Three reviewers raised a concern regarding the novelty of our paper. We lay out the reasons here why we believe our work is novel, and not incremental.

To our knowledge, this is the ***first*** work that demonstrates how to:

(1) **determine** and (2) **continuously adjust** the memory containers (how many bits should be used when storing floating-point mantissas and exponents in memory), and to do so (3) **on-the-fly**, for the purpose of (4) making **training** *itself* faster and/or more energy efficient.
If this is an incorrect statement, please help with a reference. We have been scouring for published works and nothing comes up.

Reviewers referred to two classes of past works as evidence that this work lacks novelty. Despite these being training methods that do adjust datatypes, they are fundamentally different in their goal and in their effect on training time and energy efficiency:

1. Training methods that use mixed or reduced datatypes require hand selecting those in advance and do not adapt them nor learn them. We instead continuously learn and adapt them, without being stuck with a few predetermined datatypes. We also do not change the computation arithmetic (only the values).

2. Training for reduced precision inference optimizes the datatype that *inference* will use. They do not change how the values are stored and read from memory during training. These often increase training time and energy since they target inference instead. Furthermore, these methods cannot be readily adapted to improve training. We have discussed this in the introduction of our paper as well.

If one keeps going towards lower-levels and focuses on individual pieces in isolation, similarities may seem to emerge (e.g., an augmented loss function, or a differentiable precision function), but this is missing the forest from the trees. We build on the collective experience by adopting and adapting components, introducing new ones, and combining them together into a novel technique that solves a problem that so far has not been addressed.

In summary, past work does not solve the problem we are solving. It either does not adapt and uses preselected datatypes, or, while it does improve inference, produces fatal flaws that can’t be readily removed, making it unsuitable for improving training.

---

### Author Response · Authors · 2022-11-19
**Post Review Comment**

Dear Reviewers, we hope you are doing well and thank you again for you time and comments! We tried to incorporate them into our revision. The major changes are:

1) Further discussion of effects of Quantum Mantissa in Appendix C, starting at page 18. Here we discuss the effects of the new hyperparameter gamma, consistency of results over different training runs, early stopping to reduce overheads and results on different tasks
2) Further discussion of effects of BitChop in Appendix D, starting at page 21. Here we discuss the effect of the exponential decay factor (alpha) used in the moving average, the variation in accuracy across different runs and the effect of the threshold for the change in the moving average.
3) Expanding our results in Section 3 to 2 NLP models (BERT and GPT-2), 1 recommendation model (DLRM) and ResNet50 on ImageNet
4) Small changes to clarify the text and fix some issues

We believe that your comments and the newly presented data improve the quality of the draft. If you have any more comments, questions or concerns we would be happy to answer them.

---

### Decision · Program_Chairs · 2023-01-20

**Decision:**

Reject

**Justification For Why Not Higher Score:**

Paper is lacking in novelty and evaluation. While reviewers include NLP fine tuning results during discussion phase and the reviewer saw it a step in the right direction, but it is still insufficient to convince that author's approach is ready to replace existing mixed-precision approach for NLP pretraining.

After reviewer discussion with authors, some concerns were addressed but still many raised weakness weren't fully addressed. In the end,  no reviewer saw the submission meeting the bar for ICLR publication.


**Justification For Why Not Lower Score:**

N/A

**Metareview: Summary, Strengths And Weaknesses:**

The paper is about utilizing  reduced-precision quantization methods for efficient deep neural network training. The authors propose a combination of Gecko, Quantum Mantissa and BitChop hardware-accelerated techniques for optimizing the memory footprint of neural network model training through low-precision floating point tensors. In evaluation, authors show memory saving and negligible accuracy drop for image classification models.

Here quoting the strengths and weakness highlighted by the reviewers

Strengths

- Problem is well-motivated and the problem chosen is timely since scaling model size is a recent trend in deep learning. The memory footprint of model training is an important problem for training efficiency and costs
- Dynamic and adaptive techniques are likely most effective because of high variability of models, hardware, and training phases.
- Hardware-acceleration is probably the most efficient way to realize these optimization techniques.

Weaknesses

- Only consideration of image models based on CNNs. Interest in mixed-precision training is mostly driven by NLP models which are lacking.
- The paper does not discuss the handling of overflows/underflows during training, which are important issues for mixed-precision training in practice.
- The opportunity for low-precision representation of training tensors because of their value distribution is not a novel observation and is well-studied. The paper's contributions may be considered incremental rather than significant.
- No comparison to integer quantized neural networks
- No code for reproducibility
- In depth analysis is lacking in terms of training convergence and optimization of bit-width for the mantissa.
- Proposed three quantization techniques seem to be separately proposed with incremental ideas, limiting the novelty of this paper
- Claimed advantages of the proposed hardware are hard to validate, since there is no careful analysis of the proposed architecture and the performance comparison with the state-of-the-art accelerators.

There are mixed evaluations on readability of the paper but the majority had issues on clarity.